# Automatic Stage Lighting Control: Is it a Rule-Driven Process or Generative Task?

**Zijian Zhao**[1]**, Dian Jin**[2]**, Zijing Zhou**[3]**, Xiaoyu Zhang**[4] [*]
[1]The Hong Kong University of Science and Technology [2]The Hong Kong Polytechnic University
[3]The University of Hong Kong [4]City University of Hong Kong

## Abstract

Stage lighting is a vital component in live music performances, shaping an engaging experience for both musicians and audiences. In recent years, Automatic Stage Lighting Control (ASLC) has attracted growing interest due to the high costs of hiring or training professional lighting engineers. However, most existing ASLC solutions only classify music into limited categories and map them to predefined light patterns, resulting in formulaic and monotonous outcomes that lack rationality. To address this gap, this paper presents Skip-BART, an end-to-end model that directly learns from experienced lighting engineers and predict vivid, human-like stage lighting. To the best of our knowledge, this is the first work to conceptualize ASLC as a generative task rather than merely a classification problem. Our method adapts the BART model to take audio music as input and produce light hue and value (intensity) as output, incorporating a novel skip connection mechanism to enhance the relationship between music and light within the frame grid. To address the lack of available datasets, we create the first stage lighting dataset, along with several pre-training and transfer learning techniques to improve model training with limited data. We validate our method through both quantitative analysis and an human evaluation, demonstrating that Skip-BART outperforms conventional rule-based methods across all evaluation metrics and shows only a limited gap compared to real lighting engineers. The self-collected dataset, code, and trained model parameters of this paper are provided at `https://github.com/RS2002/Skip-BART`.

## 1 Introduction

Recently, multi-modal machine learning has significantly advanced various fields in Music Information Retrieval (MIR), including video-to-music generation (Tian et al., 2024), music video (MV) generation (Chen et al., 2025), and lyrics generation (Ma et al., 2024). Among these techniques, Automatic Stage Lighting Control (ASLC) [1] shows potential in live performances by shaping the atmosphere and influencing the emotions of both musicians and audiences, while offering an affordable cost.

Currently, most existing ASLC methods are based on predefined rules that classify music into finite categories with deterministic patterns. The style-based (Stanescu et al., 2018) and emotion-based (Hsiao et al., 2017) classification methods are two mainstream approaches. However, these methods have several limitations: First, due to limited datasets and labeling difficulty, the accuracy of classification models is often insufficient, and the defined categories are typically coarse-grained. Besides the biases caused by misclassification, this coarse-grained labeling directly impacts model performance. For example, the atmosphere black metal and folk black metal are entirely different, yet they may be grouped with thrash metal under the metal category, or even combined with punk in the broader rock category. Second, as lighting is viewed as an amalgam of technology and art (Gillette & McNamara, 2019), it is debatable whether we can simply use certain patterns to define it. Additionally, McDonald et al. (2022) have demonstrated that limited relationship between hue

---

[*]Corresponding Author: Xiaoyu Zhang (xiaoyu.zhang@cityu.edu.hk)

[1]To avoid confusion, in this paper, "lighting control" and "lighting generation" are used interchangeably.

of light and emotion, raising questions about the appropriateness of mapping emotions solely to the light.

To address this research gap, we propose to treat ASLC as an art content generation task rather than merely a mechanized mapping process. Inspired by the success of generative models in MIR, we develop an end-to-end deep learning methodology that trains a network directly on ground truth data from professional lighting engineers. Initially, we introduce an automatic label generation method from video data and create a self-collected dataset titled Rock, Punk, Metal, and Core - Livehouse Lighting (RPMC-L$^2$) to address the scarcity of training datasets in this field. Subsequently, we present a carefully designed neural network called Skip-Bidirectional Auto-Regressive Transformers (Skip-BART) for stage lighting generation from music. We also incorporate transfer learning and a pre-training method to enable the model to learn effectively from limited data. We evaluate our methodology through both quantitative analysis and human evaluation. The results indicate that our method outperforms conventional rule-based approaches across all metrics and achieves performance comparable to that of human lighting engineers. In summary, the main contributions of this paper are as follows:

- To the best of our knowledge, we are the first to frame ASLC as a generation task rather than a rule-driven mapping process, based on the intuition that ASLC is essentially an art creation process for humans. To support our claim, we study the offline primary ASLC task, which is modeled by taking a music sequence as input and generating a single light hue and value as output. We further evaluate this model on a real-world dataset, demonstrating the superiority of generative methods over previous rule-based approaches. Our novel perspective on generation provides a strong foundation for future research in ASLC.

- We develop Skip-BART, a novel end-to-end deep learning framework to address the aforementioned generative task for ASLC. Building on BART, our approach incorporates adapted embedding and head layers to support both music and light modalities, as well as a novel skip-connection module to enhance the relationship between music and light within a fine-grained context. Additionally, we introduce pre-training and transfer learning mechanisms to improve model performance under limited training data. During inference, we implement a Restricted Stochastic Temperature-Controlled (RSTC) sampling method to ensure both diversity and stability in the generated results.

- To validate our method, we present the first stage lighting dataset, RPMC-$L^2$, and train Skip-BART on it. Through quantitative analysis and human evaluation, we demonstrate that our method outperforms conventional rule-based approaches across all metrics. Specifically, our method yields a significant difference ($p < 0.001$) compared to conventional rule-based methods, while showing no significant difference compared to human lighting engineers ($p = 0.72$) in the human evaluation. This suggests that our proposed method closely matches human lighting engineering performance. Furthermore, our method achieves promising results in cross-domain scenarios with other music styles, significantly outperforming previous rule-based methods.

## 2 RELATED WORK

Although ASLC is not a widely explored topic in MIR, it has received some attention in previous works. Most prior methods can be summarized as a paradigm that first classifies music into predefined categories, such as emotion and style, and then defines a corresponding light pattern for each category. For instance, works (Stanescu et al., 2018; Lei et al., 2021; Kanno & Fukuhara, 2022; Liao et al., 2023) utilize style and theme for classification based on audio or lyrics, respectively, while Mabpa et al. (2021) classify music using chords. However, these methods exhibit significant limitations. Firstly, there is insufficient theoretical support for mapping these categories to light. Additionally, coarse-grained classification severely impacts model performance. For example, Mabpa et al. (2021) classify chords into only 24 major and minor chords, neglecting more nuanced chords, such as augmented, seventh, and suspended chords, which possess distinct colors and functions. Furthermore, many pieces of music lack lyrics or contain lyrics that are difficult to recognize (such as screaming in heavy music), rendering the method of Liao et al. (2023) ineffective.

Additionally, a series of works focus on emotion classification (Bonde et al., 2018; Hsiao et al., 2017; Moon et al., 2015), employing various machine learning methods, including Support Vector

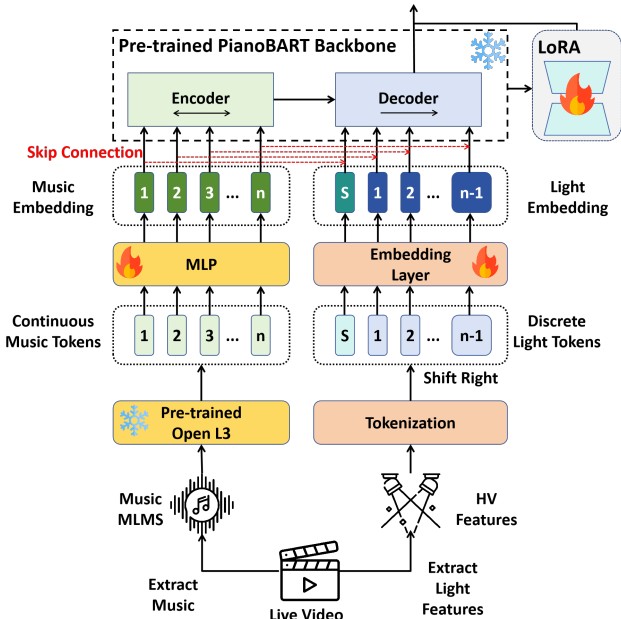

Figure 1: Network Architecture: In the figure, 'ice' represents the frozen parameters, and 'fire' denotes the trainable ones.

Machines (SVM), Support Vector Regression (SVR), and Multi-Layer Perceptrons (MLP). Vryzas et al. (2017) extend this to the field of drama. However, the relationship between music or performance emotion and light remains controversial, with different studies presenting varying perspectives (Kanno & Fukuhara, 2022; McDonald et al., 2022). We argue that simply mapping emotion to light is not a viable approach for two reasons: (i) Although Kanno & Fukuhara (2022) demonstrate that certain light colors are frequently associated with specific emotions by professional lighting engineers, this does not imply that these colors must be used when conveying those emotions. For example, in the dataset associated with this paper, we observe that the green hue appears in groove metal associated with anger emotion, contradicting the color theory suggesting red should be used. (ii) Beyond the granularity issue mentioned earlier, the performance of all rule-based methods depends on the accuracy of the classifier. In some works (Zhao, 2025), the emotion classification accuracies of various methods fall below 80%, raising doubts about the practicality of these methods at their current stage.

Moreover, Tyroll et al. (2020) present a different workflow, where a feature extractor is first trained via an autoencoder, and then the feature embeddings are directly mapped to light. However, the feature extraction capacity of autoencoders has been shown to be limited (Van Den Oord et al., 2017), and the mapping from feature embeddings to light is also ambiguous. In contrast, we propose the first end-to-end method for stage lighting control, framing the task as a generation problem rather than a classification task. Compared to previous hierarchical mapping methods, our approach directly trains the model on real-world data from lighting engineers, resulting in improved performance.

## 3 METHODOLOGY

### 3.1 OVERVIEW

In this section, we introduce an end-to-end model, Skip-BART, designed to generate single principal stage lighting based on music. Similar to previous works, we focus exclusively on offline primary single lighting generation. In the following content, we describe our model structure, as depicted in Fig. 1. We modify the embedding layers to accommodate our task's input, utilize pre-trained models from other tasks for enhancement through transfer learning, and incorporate a skip-connection

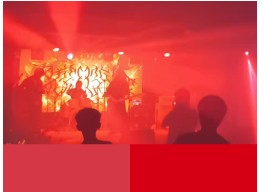 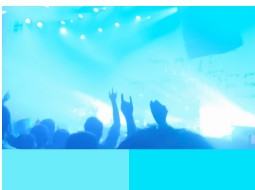 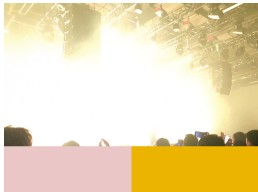 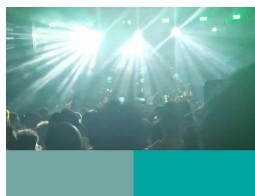

**(a) Normal Warm Light** **(b) Normal Cold Light** **(c) Extreme Warm Light** **(d) Extreme Cold light**

Figure 2: Each subfigure shows the input scene on the top, the result of the direct HSV extraction method in the bottom-left, and our extraction method in the bottom-right. **(a)-(b)** Both methods accurately extract the dominant hue. **(c)-(d)** Our method extracts colors closer to the original appearance.

mechanism to better capture the aligned relationship between music and light. Subsequently, we outline the workflow of our method, which includes MLM-based pre-training, end-to-end fine-tuning, and inference using a RSTC sampling method.

Before delving into the details of our methodology, we first provide a brief introduction to the data processing, with additional details left in Appendix C.3. To facilitate the analysis of lighting conditions, the lighting information is preprocessed in the HSV color space. The HSV color space separates the brightness (Value) information and the color (Hue) information, which makes the control and generation of lighting more intuitive, independent, and stable. Specifically, we set the Saturation (S) channel to a constant value[2] of 255 (100%) to simulate the lighting effect and counteract environmental influences, ensuring that after factors such as air scattering, fog, or surface reflections reduce color purity, the light maintains vividness and minimizes distortions like fading or whitening. Compared to the light extraction approach that only considers directly extracting HSV features, our data processing method enables more accurate color extraction under extreme intensity lighting conditions, as shown in Fig. 2.

Then, for each music sequence $x = \{x_1, x_2, \ldots, x_T\}$, we extract the corresponding light sequence $y = \{y_1, y_2, \ldots, y_T\}$, where $T$ represents the number of frames. For each frame $t$, we define $y_t = [\bar{h}_t, \bar{v}_t] \in \mathbb{Z}_*^2$, where $\bar{h}_t$ and $\bar{v}_t$ denote the hue and value extracted from the HSV space.

## 3.2 NETWORK ARCHITECTURE

The model structure is illustrated in Fig. 1. In our task, the model takes an entire piece of music as input and generates the corresponding lighting sequence. We believe that BART (Lewis, 2019) is well-suited for this task, as the bi-directional encoder effectively extracts contextual information simultaneously, while the unidirectional decoder generates outputs in an auto-regressive manner, thereby maintaining correct contextual correlations. To adapt BART for our task, we first modify its embedding layers to support the music and light input. To align the dimensions between Open L3 and BART, we use a MLP to further extract features from the embeddings produced by Open L3. The hue and value obtained from Appendix C.3 are then embedded using respective embedding layers and subsequently combined, allowing them to be input into the decoder of the backbone. As noted in (Zhao et al., 2024), the tokenization process may lead to some information loss. However, directly processing hue data with an MLP presents challenges due to the cyclical nature of hue values: the minimum and maximum hue values can be quite similar, complicating the learning process. In contrast, the embedding layer effectively captures the relationships between different tokens, making it more suitable for handling cyclic data like hue.

When applying BART for the lighting generation task, we notice that it can be challenging for the model to learn the rhythm, meaning that the light frequency and pattern changes do not always align with the music. One reason for this is that the attention mechanism in the decoder struggles to determine which light frame corresponds to which music frame. Although it is intuitive for humans to recognize the one-to-one correspondence (since the music input $x$ and light output $y$ have the same length), the model finds it difficult to learn this relationship directly from complex and noisy

---

[2]To avoid ambiguity, we use Value or V to refer to the V component in HSV, and value to represent its numerical value here.

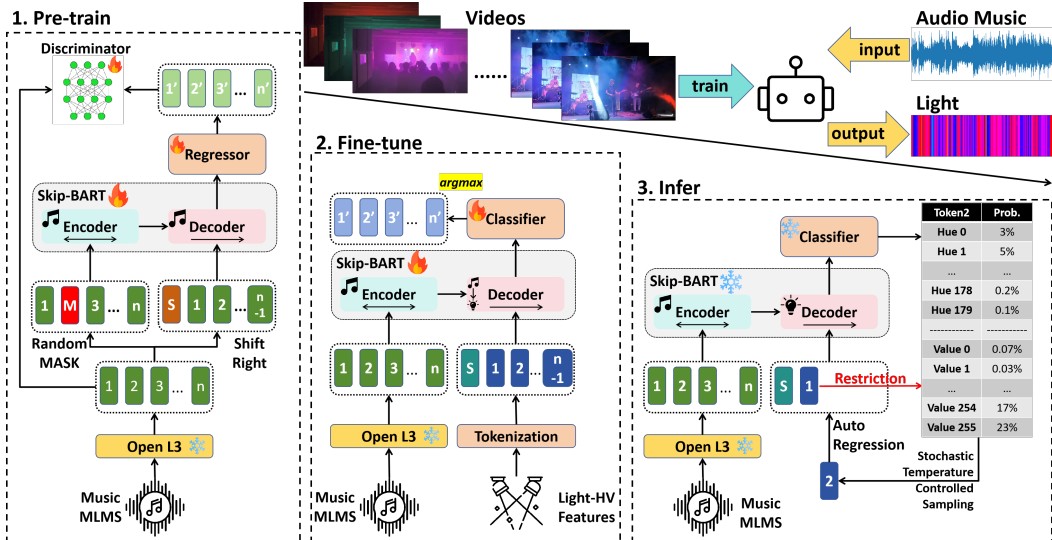

Figure 3: Workflow of Skip-BART

data in an end-to-end manner. To address this, we propose a skip mechanism that directly combines the music embedding and light embedding before inputting them to the decoder, thereby explicitly indicating the correspondence relationship. For example, the embedding of the light token $y_i$ will be combined with the embedding of the music token $x_{i-1}$ (noting that there is a one-position shift, as all inputs to the decoder should be shifted right by one position).

Additionally, transfer learning has demonstrated the benefits of using trained parameters from other domains (Pan, 2014). Given the limited datasets in MIR, we aim to introduce more external knowledge to enhance model performance. Consequently, we adopt the backbone of PianoBART (Liang et al., 2024), a large-scale pre-trained model for symbolic music understanding and generation, in our Skip-BART framework. Additionally, to embed as much knowledge as possible, we utilize the Drop And Rescale (DARE) (Yu et al., 2024) method to combine the fine-tuned parameters of PianoBART across all downstream tasks, including priming, melody extraction, velocity prediction, composer classification, and emotion classification. The combined model parameters are as follows:

$$
\theta = \theta_{pre} + \lambda \sum_i (\theta^i_{DARE} - \theta_{pre}) \,,
$$
$$
\theta^i_{DARE} = \theta_{pre} + \frac{(\theta^i - \theta_{pre}) \odot m_i}{1 - p} \,,
$$
$$
m_i \sim \text{Bernoulli}(p) \in \{0, 1\}^d \,,
$$
(1)

where $\theta$ represents the combined model parameters, $\theta_{pre}$ denotes the pre-trained model parameters, $\theta^i$ represents the fine-tuned model parameters for the $i^{th}$ downstream task, $d$ is the dimension of the parameters, $p$ is the drop rate, $\lambda$ is the scaling term, and $\odot$ represents element-wise multiplication. We then employ LoRA (Hu et al., 2022) for efficient tuning in our tasks. Due to page limitations, further details can be found in (Yu et al., 2024). Finally, for both pre-training and fine-tuning, we employ different heads for various tasks, which will be detailed in the next section.

*In summary, our proposed network is based on the BART backbone, utilizing OpenL3 to extract music audio features and a tokenization mechanism with an embedding layer to extract light features, effectively addressing the cyclic problem of hue. A novel skip-connection mechanism is introduced to enhance the relationship between music and light within the frame grid. Additionally, the backbone directly incorporates pre-trained parameters from PianoBART, providing a solid starting point for subsequent training, while using LoRA for efficient tuning.*

### 3.3 WORKFLOW

#### 3.3.1 MLM-BASED PRE-TRAINING

To fully utilize the training data, we first employ an unsupervised pre-training method that encourages the model to learn the underlying data structure. To prevent information leakage, we do not use light data during this phase. Instead, we input audio music into both the encoder and decoder. As noted in transfer learning, the parameters of the middle layers of the model are generally similar across different modalities or tasks. This similarity aids the decoder in initializing appropriate parameters, allowing knowledge to be transferred from music to light during fine-tuning.

For pre-training the model on music data, we opt for the MLM task, following the approaches in (Zhao et al., 2024; Zhu et al., 2021; Lewis, 2019). A MLP is used after the decoder's output to map the output hidden state to music sequence. We first utilize Open L3 to embed the audio music, resulting in an embedding sequence $e = \{e_1, e_2, \ldots, e_n\} \in \mathbf{R}^{n \times h}$, where $h$ is the hidden dimension. We then randomly replace $k\%$ of the tokens with [MASK] tokens and train the model to recover them. Since the music tokens are continuous, we design the [MASK] token as a random variable sampled from a normal distribution, based on (Zhao et al., 2024):

$$
\begin{aligned}
[\text{MASK}] &= [M_1, M_2, \ldots, M_h]' \in \mathbb{R}^{1 \times h} \,, \\
M_i &\sim N(\mu^i, \sigma^i) \,,
\end{aligned}
\tag{2}
$$

where $'$ represents transposition, $\mu^i$ and $\sigma^i$ denote the mean and standard deviation of sequence $e$ in the $i^{th}$ feature dimension. In this manner, the model not only aims to recover the obscured tokens but also to identify which tokens have been replaced by [MASK], further assisting the model in learning the structure of the music sequence. To enhance the realism of the recovered output, we also employ a GAN-based discriminator to distinguish between the original music and the reconstructed music, as described by (Zhao et al., 2024):

$$
\begin{aligned}
&\min_R \max_D V(D, R) \\
&= \min_R \max_D E_e[log(D(e))] + E_e[log(1 - D(R(e)))] \,,
\end{aligned}
\tag{3}
$$

where $D$ and $R$ represent the discriminator and recoverer, respectively. For the discriminator, we adopt the same architecture as the sequence classifier in Chou et al. (2021). It employs a modified self-attention layer to capture the global information from the embedding sequence $e$, followed by a linear layer that classifies the extracted feature as True or False.

Here, to enhance parameter efficiency, we allow the discriminator and recoverer to use a shared Skip-BART backbone while employing separate heads, as suggested by (Zhao et al., 2024). Thus, we can express the pre-training loss function of Skip-BART as follows:

$$
\begin{aligned}
L_{pre} &= \alpha_1 l_1 + \alpha_2 l_2 + \alpha_3 l_3 \,, \\
l_1 &= \text{MSE}(\hat{e}, e) \,, \\
l_2 &= \text{MSE}_{i \in M}(\hat{e}_i, e_i) \,, \\
l_3 &= \text{CrossEntropy}(D(\hat{e}), 1) \,,
\end{aligned}
\tag{4}
$$

where $M$ represents the token index set of those replaced by [MASK], $\hat{e}$ denote the outputs of Skip-BART, $l_1$ is the reconstruction loss resembling that of an autoencoder, $l_2$ focuses on the recovery loss for the masked tokens, $l_3$ is the realism loss to enhance the believability of the recovered output, and the parameters $\alpha_1, \alpha_2, \alpha_3$ are weights. Additionally, the loss function for the discriminator can be defined as:

$$
L_{dis} = \text{CrossEntropy}(D(\hat{e}), 0) + \text{CrossEntropy}(D(e), 1) \,.
\tag{5}
$$

Due to page limitations, we have included additional details regarding pre-training in Appendix A.

#### 3.3.2 END-TO-END FINE-TUNING

During fine-tuning, we utilize the Language Model (LM) task (Radford et al., 2018) (next token prediction) to supervise the model in generating the corresponding lighting sequences. We frame this task as a classification problem where the model needs to classify the hue and value at each token

position. The classification vocabularies are defined by the binning scheme used in the histograms $D_{h,t}$ and $D_{v,t}$, as described in Eq. 11. Consequently, we employ two MLPs to map the decoder output to the classification probabilities for hue and value, respectively. The loss function can be expressed as:

$$L_{stf} = \beta_1 \text{CrossEntropy}(\hat{h}, \overline{h}) + \beta_2 \text{CrossEntropy}(\hat{v}, \overline{v}) . \tag{6}$$

where $\hat{h}$ and $\hat{v}$ denote the model outputs, while $\beta_1$ and $\beta_2$ are hyper-parameters. We observe that the learning speeds for these two attributes are not the same. To encourage the model to focus more on the attributes that are learned more slowly, we use adaptive weights as proposed in (Zhao, 2025):

$$\beta_i^j = \frac{\frac{1}{acc_i^{j-1}}}{\sum_{i=1}^2 \frac{1}{acc_i^{j-1}}} , \tag{7}$$

where $\beta_i^j$ represents the weight in epoch $j$, and $acc_1^{j-1}$ and $acc_2^{j-1}$ denote the model accuracy for hue and value, respectively, in the validation set of the $j^{th}$ epoch.

### 3.3.3 RSTC-BASED INFERENCE

During the inference phase, Skip-BART generates the lighting sequence in an auto-regressive manner. To prevent model degradation and to implement a stochastic temperature-controlled sampling method (Hsiao et al., 2021), we represent this as follows:

$$\hat{y}_i \sim \text{softmax} \left( \frac{\textbf{Skip-BART}(x, [\hat{y}_1, \ldots, \hat{y}_{i-1}])}{t} \right) , \tag{8}$$

where $t$ represents the temperature, and $[\hat{y}_1, \ldots, \hat{y}_{i-1}]$ denotes the previously generated lighting sequence. The right term of Eq. 8 consists of two vectors, representing the probabilities of each category for hue and value, respectively. The hue and value are then sampled from these two probability vectors.

Additionally, to avoid overshooting, we ensure that the successive two lighting tokens do not exceed a specified threshold:

$$\begin{cases} \mathcal{D}_h(\hat{h}_i, \hat{h}_{i-1}) & = \min\{|\hat{h}_i - \hat{h}_{i-1}|, 180 - |\hat{h}_i - \hat{h}_{i-1}|\} < d_h^t , \\ \mathcal{D}_v(\hat{v}_i, \hat{v}_{i-1}) & = |\hat{v}_i - \hat{v}_{i-1}| < d_v^t , \end{cases} \tag{9}$$

where $\mathcal{D}(\cdot, \cdot)$ is the distance function $d^t$ is the threshold. To satisfy this restriction, we set the probability of all categories over this distance to zero during stochastic temperature-controlled sampling. We refer to this method as Restricted Stochastic Temperature-Controlled (RSTC).

## 4 EVALUATION

### 4.1 SETUP

To illustrate the effectiveness of our method, we compare it with other approaches through both quantitative analysis and an human evaluation. In the quantitative analysis, we evaluate the differences among our method, a conventional rule-based method, and several ablation studies in relation to the ground truth. We assume that the closer the generated results are to the ground truth, the better the model performs, a hypothesis that will be validated through human studies. During the human evaluation, participants are asked to score various methods, including the proposed Skip-BART, the Ablation Study, the previous methods (referred to as the Rule-based Method), and the ground truth across different dimensions.

More details about the model configuration and training process can be found in the Appendix B. The experiments were conducted using the PyTorch framework (Paszke et al., 1912) on a server running Ubuntu 22.04.5 LTS, equipped with an Intel(R) Xeon(R) Gold 6133 CPU @ 2.50 GHz, two NVIDIA 4090 GPUs, and one NVIDIA A100 GPU.

Moreover, as mentioned in introduction, there is currently no dataset available for the lighting generation task, which is one reason previous methods have relied on rules. We address this gap by

Table 1: Quantitative Results: The best result is indicated in **bold**, and the second best is indicated by underline. This notation will remain consistent in the following tables.

| Methods | RMSE↓ | | MAE↓ | | $\text{corr}(|\Delta|)(\times 10^{-2})$↑ | |
| --- | --- | --- | --- | --- | --- | --- |
| | **Hue** | **Value** | **Hue** | **Value** | **Hue** | **Value** |
| **Rule-based** | 48.67 | 93.39 | 43.43 | 86.55 | 0.50 | 0.58 |
| **Skip-BART** | **36.13** | **60.74** | **28.72** | **51.27** | 0.88 | **2.94** |
| **w/o skip connection** | 36.89 | 68.33 | 29.44 | 58.34 | **1.15** | 0.30 |
| **w/o light embedding** | 51.04 | 67.25 | 41.50 | 54.87 | 0.80 | 0.70 |
| **train from scratch** | 36.63 | 67.49 | 28.83 | 57.22 | 0.69 | 0.53 |
| **pre-train w/o random [MASK]** | 49.97 | 64.45 | 42.07 | 52.63 | 0.54 | 1.11 |
| **pre-train w/o discriminator** | 50.40 | 68.09 | 41.52 | 56.54 | 0.46 | 1.13 |

proposing the first stage lighting dataset named 'RPMC-L$^2$' (Rock, Punk, Metal, and Core - Live-house Lighting). This dataset is highly diverse, collected by the authors over the past five years, including contributions from numerous bands, livehouses, and four different capturing devices, covering a wide range of styles such as metalcore, alternative rock, and post-punk. We construct the videos using the proposed method in Appendix C.3 from raw footage. Our dataset includes 699 samples, with lengths ranging from 20 seconds to around 5 minutes, making it a relatively medium-sized dataset in MIR. More details can be found in the Appendix C.

## 4.2 QUANTITATIVE ANALYSIS

In the quantitative experiment, we compare the generation results of different methods against the ground truth. Since our method utilizes random sampling during training, comparing loss and accuracy has limited significance, similar to most generative works (Liang et al., 2024). Therefore, we directly compute four metrics between the generated results and the ground truth: (1) Root Mean Square Error (RMSE) and (2) Mean Absolute Error (MAE), both measuring absolute prediction errors. For the Hue channel, RMSE/MAE are computed on the circular angle differences, ensuring that values like 0° and 179° are treated as close rather than far apart, as shown in Eq. 9. (3) $\text{corr}(|\Delta|)$ measures the consistency of change magnitudes by computing the Pearson correlation of the absolute first-order differences between predictions and ground truth. The final value is multiplied by 100. The results are presented in Table 1, where it is evident that our method outperforms all ablation studies and the rule-based method in both hue and value, achieving the lowest error. Since the rule-based method does not directly learn from the ground truth, it may introduce some unfairness. We provide further explanations in the human evaluation section.

## 4.3 HUMAN EVALUATION

To better assess how our results align with human preferences, we designed a questionnaire to collect participants' feedback on the stage lighting effects created by each method. We recruited 38 respondents through social media platforms and live music venues, with the human evaluation spanning half a month. The participants included 20 males and 18 females aged 18–55 (predominantly 18–30), among whom 9 had professional experience in lighting design, music production, or stage art, and over half were frequent live music attendees. All participants reported normal hearing, normal or corrected-to-normal vision, and no history of color blindness or color weakness. All participants are required to evaluate three music pieces alongside the lights across six metrics (Erdmann et al., 2025), with each metric scored from 1 to 7 (the higher the score, the better the evaluation). The lighting conditions were derived from four sources: Ground Truth, Skip-BART, Rule-based Method, and Skip-BART (w/o skip connection). *For the ablation study, we specifically evaluate Skip-BART without the skip connection. We considered two main factors: first, the efficiency of these components has been widely demonstrated in other fields; second, increasing the number of comparative methods complicates data collection and may discourage user participation, as longer questionnaires can lead to fatigue and confusion while viewing the same lighting video for a particular piece of music. For simplicity, we will refer to Skip-BART (w/o skip connection) as the Ablation Study in this section and the following tables.* Due to page limitations, we outline the primary experimental

process and results below, while the Appendix E contains the complete survey questionnaire and the analytical results.

We first calculated mean scores for each video across all three music backgrounds. These averaged values served as composite scores for subsequent statistical analysis. Then, we employed Repeated Measures ANOVA to examine the scores of four videos across overall and six individual metrics, while controlling for within-subject variability inherent in repeated measurements. To control Type I errors in multiple comparisons, we applied the Bonferroni correction. For method-wise differentiation, we conducted post hoc comparisons using Bonferroni-corrected paired t-tests.

As shown in Table 2 and Table 3, Ground Truth receives the highest overall score (M = 4.5, SD = 0.9), closely followed by our Skip-BART(M=4.3, SD=0.9) and the Skip-BART (w/o skip connection)(M=4.1, SD=0.8), both of which are near the GT level. All three methods significantly outperform the rule-based approach(M=2.7, SD=1.3,p<0.05), a trend that holds for all the metrics. The only metric where Ground Truth does not achieve the best score is in emotion matching, supporting the perspective that the emotional aspects of music may have a limited connection to lighting control (McDonald et al., 2022). We also observe that for the rule-based method, the score of the emotion metrics(M=3.1, SD=1.5) is higher than that of other metrics, which validates our reproduction efforts.

Table 2: Human Evaluation Scores: The six metric scores and overall evaluations of the four objects are expressed as Mean ± Standard Deviation (M ± SD) . The **bold** text represents the best result, while the underlined text indicates the second-best result.

| Method | Emotion | Impact | Rhythm | Smoothness | Atmosphere | Surprise | Overall |
|--------|---------|--------|--------|------------|------------|----------|---------|
| | | | *In-domain Evaluation* | | | | |
| **Ground Truth** | 4.50±0.93 | **4.48±0.99** | **4.61±0.99** | **4.62±1.07** | **4.49±0.89** | **4.34±1.10** | **4.51±0.88** |
| **Skip-BART** | **4.69±0.87** | 4.39±0.95 | 4.50±1.06 | 4.32±1.12 | 4.32±0.93 | 3.83±1.06 | 4.35±0.87 |
| **Ablation Study** | 4.31±0.94 | 3.78±0.96 | 4.54±1.08 | 4.43±1.12 | 4.11±0.98 | 3.50±1.00 | 4.11±0.84 |
| **Rule-based** | 3.12±1.52 | 2.65±1.39 | 2.54±1.47 | 2.56±1.27 | 2.77±1.50 | 2.35±1.40 | 2.67±1.29 |
| | | | *Cross-domain Evaluation* | | | | |
| **Skip-BART** | **4.31±1.05** | **4.18±1.09** | **4.60±1.14** | **4.61±0.92** | **4.41±1.09** | **3.91±1.17** | **4.34±0.92** |
| **Ablation Study** | 3.94±1.31 | 3.93±1.12 | 4.44±1.02 | 4.42±1.05 | 4.04±1.11 | 3.79±1.15 | 4.10±0.97 |
| **Rule-based** | 3.61±1.31 | 3.04±1.20 | 3.21±1.31 | 3.88±1.15 | 3.34±1.19 | 2.96±1.41 | 3.34±1.13 |

Comparing our Skip-BART with the Skip-BART without skip connection, from which we removed the skip connection, we find that our method consistently outperforms in all metrics except for rhythm($\Delta M$ = -0.035, SD=0.12, p=1.00) and smoothness($\Delta M$ = -0.11, SD=0.16, p=1.00), with minimal performance gaps. The small gap in rhythm can be attributed to randomness. In terms of smoothness, the skip connection links each frame of music to light directly, which might cause significant changes in light during major transitions in the music. Overall, our method greatly outperforms conventional rule-based methods and achieves performance comparable to that of professional lighting engineers.

Table 3: Pairwise Comparisons of Overall Scores for the Four Objects: The symbols * and ** denote that the significance level (p) for the difference in means ($\Delta M$) is less than 0.05 or 0.01, respectively.

| Comparison | In-domain Evaluation | | | Cross-domain Evaluation | | |
|------------|:---:|:---:|:---:|:---:|:---:|:---:|
| | $\Delta M$ | SD | p | $\Delta M$ | SD | p |
| **Ground Truth vs. Skip-BART** | 0.16 | 0.10 | 0.724 | – | – | – |
| **Ground Truth vs. Ablation Study** | 0.40 | 0.10 | **0.003**** | – | – | – |
| **Ground Truth vs. Rule-based** | 1.84 | 0.21 | **<0.001**** | – | – | – |
| **Skip-BART vs. Ablation Study** | 0.23 | 0.10 | 0.152 | 0.24 | 0.12 | 0.167 |
| **Skip-BART vs. Rule-based** | 1.68 | 0.19 | **<0.001**** | 1.00 | 0.17 | **<0.001**** |
| **Ablation Study vs. Rule-based** | 1.44 | 0.16 | **<0.001**** | 0.76 | 0.15 | **<0.001**** |

Additionally, to evaluate our model's performance in a cross-domain scenario, we compare different methods across various music styles. For fairness, we utilize songs generated by Suno (Suno) instead

of selecting them manually. This approach allows us to assess the model's performance on entirely novel songs. Furthermore, since no ground truth is available, we omit this object and ask 30 users to evaluate other three objects across the following music genres: folk, R&B, and jazz. The detailed results are presented in Tables 2 and 3. Our proposed method consistently achieves the highest scores across all metrics, demonstrating significant differences compared to the rule-based method.

## 5 LIMITATIONS AND FUTURE WORK

While our framework demonstrates positive results, there remain several limitations and opportunities for further improvement. Below we highlight several key directions worth further exploration:

- **Generalizability Across Music Genres:** In our experiment, we have demonstrated promising results from our method on both in-domain music genres (e.g., rock, punk, metal, and hardcore) and cross-domain genres (e.g., folk, R&B, and jazz). However, further evaluation across a wider range of styles could be explored in future research. In practice, our method is independent of the dataset, allowing users to retrain or fine-tune our model based on their specific datasets. However, the scarcity of large, high-quality datasets has long posed a challenge in the MIR community. In addition to the pre-training and transfer learning mechanisms used in this paper, unsupervised domain alignment methods also present a potential paradigm, requiring only the music in the target domain without the need for light ground truth.

- **Alternative Generation Technology Exploration**: This paper introduces a generative model for lighting control, framing it as a creative task. However, as discussed in Appendix D, the model occasionally shows overly strong local fluctuations, which suggest that other popular generative paradigms and technologies deserve exploration. Additionally, emerging techniques such as Reinforcement Learning from Human Feedback (RLHF) (Ouyang et al., 2022) and transferring from general music foundation models (Suno; Yuan et al., 2025) may provide promising directions for enhancing model capability.

- **Towards More Practical Application Scenarios:** Similar to most previous studies, our method currently supports only offline primary lighting generation. Future research could explore real-time and multi-light control. However, these upgrades face significant challenges. For multi-stage control, the heterogeneity among venues must be considered, as different livehouses often have varying light positions and quantities. Regarding online control, the primary challenge is the need to predict changes in music quickly and accurately, as light control decisions must be made within seconds. However, accurately predicting a song's progression is difficult due to its artistic nature, where unexpected elements contribute to the charm of music.

- **Richer Modality Incorporation**: In our model, it is also possible to incorporate richer modalities, such as rhythm, musical sections (like verses and choruses), and lyrics. Models such as beat tracking (Chang & Su, 2024) and melody extraction (section classification) (Zhao, 2025) can also be integrated to provide more informative signals.

## 6 CONCLUSION

In this paper, we propose the first end-to-end deep learning method, Skip-BART, for stage lighting generation. We modify the BART model to enable the encoder to extract music features in a single pass and the decoder to generate light hue and intensity auto-regressively, framing ASLC as a generative task. A skip connection mechanism is introduced to enhance the relationship between corresponding music and light frames. Additionally, we incorporate effective deep learning techniques to allow the model to learn from limited data, including transfer learning, pre-training, and RSTC sampling. To support research in this field, we present a dataset construction method based on raw video and release the first stage lighting dataset, RPMC-L$^2$. Through quantitative analysis and human studies, we demonstrate the efficiency of our method and highlight the limitations of conventional rule-based approaches. Our Skip-BART shows performance closely matching that of real human lighting engineers. The results validate our perspective that music lighting control resembles an art generation task rather than a simple classification and mapping task based on predefined rules. This study aims to provide deeper insights and support for future research in this area.

## ACKNOWLEDGEMENT

We are grateful to the participants who volunteered for our human study and the anonymous reviewers who helped improve the quality of this paper. We also thank Yunzhong Luo for helping with proofreading the manuscript.

## ETHICS STATEMENT

Our study did not use any privacy data containing personally identifiable information; all information from human evaluation participants was anonymized, and only essential background data were collected for statistical and analytical purposes. At the same time, we recognize that the generated lighting may involve high flicker frequencies, strong color contrasts, or rapid brightness changes, which could pose health or discomfort risks to individuals with photosensitive epilepsy or visual sensitivities.

## REPRODUCIBILITY STATEMENT

The source code, trained parameters, and processed dataset are available at `https://github.com/RS2002/Skip-BART`

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

APPENDIX

## A  PRE-TRAINING DETAILS

In this section, we aim to provide additional details regarding the input and output, the neural network architecture, and the hyper-parameters involved in the pre-training phase, ensuring reproducibility.

During pre-training, in order to avoid any potential information leakage, we utilize only the music data without light. To provide the decoder with a robust starting point for fine-tuning, we adopt a BART-based MLM approach. In this manner, the encoder encodes the masked music sequence, while the decoder receives the shift-right ground truth as input and attempts to accurately recover the original music sequence. Although there may be a gap between the music modality in pre-training and the light modality in fine-tuning, this pre-training method has been demonstrated to be effective, as supported by the theory of cross-modal transfer learning (Niu et al., 2021), and it has been successfully applied in numerous domains (Chen et al., 2024).

Specifically, in each epoch, for a given embedding sequence $e = \{e_1, e_2, \ldots, e_n\}$, we first randomly mask some of the embeddings to generate the input for the encoder, denoted as $\tilde{e} = \{\tilde{e}_1, \tilde{e}_2, \ldots, \tilde{e}_n\}$. If $\tilde{e}_i$ is chosen, it is replaced with a random [MASK] token. Otherwise, it retains its original value $e_i$. For the decoder, we use the shift-right ground truth as input, $\{[SOS], e_1, e_2, \ldots, e_{n-1}\}$, which outputs the hidden features $f = \{f_1, f_2, \ldots, f_n\}$. These features are then fed into a regressor (a simple three-layer MLP with LeakyReLU as the activation function) to generate the estimated input sequence, $\hat{e} = \{\hat{e}_1, \hat{e}_2, \ldots, \hat{e}_n\}$. Additionally, following the recommendations of Zhao et al. (2024), we introduce a discriminator to enhance the realism of the recovered music sequence. The discriminator takes the original music sequence $e$ and the recovered sequence $\hat{e}$ as input, attempting to differentiate between them, while the decoder works to obfuscate this distinction. For the structure of the discriminator, we utilize a sequence-level classifier from MidiBERT (Chou et al., 2021), which employs a modified self-attention mechanism:

$$
\begin{aligned}
M_{\text{attn}} &= \text{Softmax}(\text{LinearLayer}(e)) \in \mathbb{R}^{n \times d}_+ , \\
z &= \text{Flatten}(M^T_{\text{attn}} \cdot e) \in \mathbb{R}^{hd} , \\
y &= \text{Softmax}(\text{LinearLayer}(z)) \in \mathbb{R}^2_+ ,
\end{aligned}
\tag{10}
$$

where $M_{\text{attn}}$ is the attention matrix, $h$ represents the feature dimension of the input, $d$ is the hidden dimension, and $y$ yields a binary classification result.

Lastly, regarding the hyperparameters, we set the mask ratio as a random number sampled from $U(0.15, 0.30)$ in each epoch. The weights $\beta_1$, $\beta_2$, and $\beta_3$ in the loss function (Eq. 4) are set to 0.8, 0.2, and 0.1, respectively.

## B  DETAILED EXPERIMENT SETUP

The experimental subjects can be summarized as follows:

- **Ground Truth (GT)**: The ground truth in our dataset is derived from real lighting engineers. In the human evaluation, we consider it as a specific method developed by humans.

- **Skip-BART**: This refers to our proposed method.

- **Ablation Study**: In comparison to conventional BART, our method's variants primarily focus on the skip connection module, the pre-training and transfer learning mechanisms, and the embedding layer. In the ablation study, we assess the model's performance without these mechanisms. We denote them as: *(i) Skip-BART (w/o skip connection)*: The model without the skip connection; *(ii) Skip-BART (from scratch)*: The model trained from scratch, without any prior knowledge from transfer learning or pre-training; *(iii) Skip-BART (w/o light embedding)*: The model that does not utilize the proposed discrete light embedding but instead employs a MLP-based continuous embedding layer for light; *(iv) Skip-BART (pre-train w/o random [MASK])*: During pre-training, the [MASK] token is fixed instead of being randomly assigned as defined in Eq. 2; *(v) Skip-BART (pre-train w/o discriminator)*: The model that does not utilize a discriminator during pre-training, i.e., $\beta_3$ in Eq. 4 is set to 0.

- **Rule-based Method**: In reproducing previous methods, we encountered several challenges, as most existing works do not release their code and parameters. Even when attempting to implement

them independently, the unreleased predefined light patterns made it impossible to map classification results to their corresponding lighting control outputs. We follow the core idea of Hsiao et al. (2017), which maps music to emotional categories and infers specific lighting patterns based on embeddings in emotional space. For emotion classification, we adopt the method of Alajanki et al. (2016), training a bidirectional Long Short-Term Memory (LSTM) model (Hochreiter & Schmidhuber, 1997) on their published dataset, DEAM. For light mapping, we utilize the approach of Nijdam (2009), taking into account both hue and value.

For our Skip-BART model, the detailed configuration in our experiment are shown in Table 4, with the backbone aligned with PianoBART (Liang et al., 2024).

Table 4: Model Configurations: This table provides a comprehensive overview of our Skip-BART settings used in our experiments.

| Configuration | Our Setting |
|---|---|
| Vocabulary Size for Hue and Value | [180, 256] |
| Lighting Embedding Dimension | 512 |
| Music Embedding Dimension | 512 |
| Input Length | 1024 |
| Number of Network Layers | 8 |
| Hidden Size | 2048 |
| Inner Linear Size | 2048 |
| Attention Heads | 8 |
| Dropout Rate | 0.1 |
| Total Number of Parameters | 240 M |
| Trainable Parameters | 19 M |
| Optimizer | AdamW |
| Learning Rate | 0.0001 |
| Batch Size | 16 |
| GPU Memory Usage | 18 GB |
| Pre-training Duration | 15 hours |
| Fine-tuning Duration | 1.5 hours |

## C  DATASET DETAILS

### C.1  OVERVIEW

The dataset was collected from 35 live performances at various commercial venues between December 2020 and July 2024 by the authors. After data cleansing, videos shorter than 20 seconds were discarded, resulting in a total of 699 retained samples. The samples primarily encompass music styles such as generalized rock, punk, metal, and core. To address copyright concerns and enhance usability, we provide extracted processed features instead of the original video files, resulting in approximately 40GB of HDF5 files.

### C.2  AUDIO PROCESSING

Various methods were employed to process the audio data, including:

- **OpenL3 (Cramer et al., 2019)**: A deep neural network architecture for extracting high-level auditory embeddings through audio-visual correspondence learning.

- **Mel Spectrogram**: Time-frequency representation employing triangular filter banks spaced according to the mel scale, approximating human auditory perception.

- **Log-Mel Spectrogram**: Decibel-scaled variant of the mel spectrogram enhancing the discernibility of low-amplitude acoustic components through logarithmic compression.

- **Constant-Q Transform (CQT)**: Time-frequency analysis technique utilizing geometrically spaced frequency bins with constant quality factors, enabling multi-resolution spectral decomposition.

- **Short-Time Fourier Transform (STFT)**: Fundamental time-localized Fourier analysis employing a sliding window function for stationary signal approximation.

- **Mel-Frequency Cepstral Coefficients (MFCCs)**: Cepstral representation derived from mel-scale filter bank energies, optimized for speech recognition through deconvolution of spectral components.

- **Chroma STFT**: Pitch-class energy profile computed from STFT magnitudes, representing harmonic content in the standard equal-tempered scale.

- **Chroma CQT**: Enhanced chroma representation leveraging the frequency resolution advantages of CQT for improved musical note analysis.

- **Chroma CENS**: Energy-normalized chroma variant incorporating temporal smoothing and logarithmic compression for robust harmonic pattern recognition.

- **Spectral Centroid**: Second-order spectral moment quantifying the center of spectral mass, correlating with perceptual brightness.

- **Spectral Bandwidth**: Fourth-order spectral moment characterizing signal dispersion about the centroid, indicative of timbral richness.

- **Spectral Contrast**: Timbral descriptor quantifying the dynamic range between spectral peaks and troughs across sub-bands.

- **Spectral Rolloff**: Cutoff frequency containing 95% of spectral energy distribution, serving as a harmonicity indicator.

- **Zero-Crossing Rate (ZCR)**: Temporal-domain measure of signal oscillation rate, particularly effective for discriminating voiced and unvoiced speech segments and percussive transients.

The respective feature dimensions are detailed in Table 5.

Table 5: Audio Features Dimension: $L$ refers to the sequence length of audio.

| Feature | Dimension |
|---|---|
| OpenL3 | ( 512, $L$) |
| Mel Spectrogram | ( 128, $L$) |
| Mel Spectrogram (dB) | ( 128, $L$) |
| CQT | ( 84, $L$) |
| STFT | (1025, $L$) |
| MFCC | ( 128, $L$) |
| Chroma Stft | ( 12, $L$) |
| Chroma Cqt | ( 12, $L$) |
| Chroma Cens | ( 12, $L$) |
| Spectral Centroids | ( 1, $L$) |
| Spectral Bandwidth | ( 1, $L$) |
| Spectral Contrast | ( 7, $L$) |
| Sspectral Rolloff | ( 1, $L$) |
| Zero Crossing Rate | ( 1, $L$) |

### C.3 LIGHTING PROCESSING

To facilitate the analysis of lighting conditions, the lighting information is preprocessed in the HSV color space. The HSV color space separates the brightness (Value) information and the color (Hue) information, which makes the control and generation of lighting more intuitive, independent, and stable. Specifically, we set the Saturation (S) channel to a constant value[3] of 255 (100%) to simulate

---

[3] To avoid ambiguity, we use Value or V to refer to the V component in HSV, and value to represent its numerical value here.

the lighting effect and counteract environmental influences commonly encountered in livehouse settings, ensuring that after factors such as air scattering, fog, or surface reflections reduce color purity, the light maintains vividness and minimizes distortions like fading or whitening. The Hue (H) and Value (V) channels are extracted frame-by-frame from the video, with H ranging from [0, 179] (H is cyclic) and V ranging from [0, 255]. For each frame, we compute the distributions (number of pixels for each value) of H and V. To avoid Hue distortion caused by low light intensity, H is extracted only from pixels where V exceeds specified thresholds, set to values in the range $\{0, 30, 60, 90, 120, 150, 180, 210, 240\}$. Users can select the threshold values that best suit their specific applications.

For this study, in order to obtain the audio sequences $s_a$, we segment it into overlapping frames $\{x_1, x_2, \ldots, x_T\}$ with a window length of 1 second and a hop size of 0.1 second. To extract the primary light from each image, we first convert each frame into the HSV color space. After flattening, each frame yields a set of $(h, s, v)$ tuples corresponding to its pixels. As described previously, we fix the saturation component $s$, and therefore omit it in subsequent description. We focus on the hue and value components and compute the histogram representations $H_{h,t} \in \mathbb{N}^{180}$ and $H_{v,t} \in \mathbb{N}^{256}$ for frame $t$ as follows:

$$
\begin{aligned}
(H_{h,t})_m &= \sum_{k=1}^{K} \mathbf{1}\{v_t^k \geq v'\} \cdot \mathbf{1}\{h_t^k = m\}, \quad m = 0, \ldots, 179, \\
(H_{v,t})_n &= \sum_{k=1}^{K} \mathbf{1}\{v_t^k \geq v'\} \cdot \mathbf{1}\{v_t^k = n\}, \quad n = 0, \ldots, 255.
\end{aligned}
\tag{11}
$$

where $\mathbf{1}\{\cdot\}$ denotes the indicator function, $h_t^k \in \{0, \ldots, 179\}$ and $v_t^k \in \{0, \ldots, 255\}$ are the hue and value of pixel $k$ in frame $t$, $K$ is the number of pixels in frame $t$, and $v' \in \{0, \ldots, 255\}$ is a threshold used to exclude low-value pixels.

We define the tokenized hue as the mode of the hue distribution, and the tokenized value as the weighted average of the value distribution for frame $t$:

$$
y_t = [\overline{h_t}, \overline{v_t}] \in \mathbb{Z}_*^2, \quad \text{where} \quad
\begin{cases}
\overline{h_t} = \text{Mode}(H_{h,t}) \\
\overline{v_t} = \text{Round}\left(\dfrac{\sum_{n=0}^{255}(n+1) \cdot (H_{v,t})_n}{\sum_{n=0}^{255}(H_{v,t})_n}\right)
\end{cases}
\tag{12}
$$

Here, we use the mean value to compute the tokenized intensity, and mode to compute the tokenized hue, as hue is a circular variable. This results in a music sequence $x = \{x_1, x_2, \ldots, x_T\}$ and the corresponding light sequence $y = \{y_1, y_2, \ldots, y_T\}$. We consider the music sequence $x$ as input and the corresponding light sequence $y$ as output.

## C.4 Overall Dataset Settings

As for the overall dataset setting for experiments, we set the sampling rate to 10 Hz, which means we obtain 10 frames for each second, and our model can process sequences up to 102.4 seconds long. For samples exceeding this length, we randomly select a 102.4-second segment from the entire sequence. We split the dataset into training, validation, and testing sets in an 8:1:1 ratio. To avoid any potential information leakage, we ensure that all shows in the testing set are not included in the training and validation sets. For pre-training, we only use samples from the training and validation sets. In the evaluation, we compare the performance of different models on the testing set.

## C.5 Multiple Hue Extraction

To support the next step of multiple light control, we first propose a multiple hue extraction method from a single frame, which can provide ground truth for future research. We recommend using a Von Mises Mixture Model (VMM) to extract the periodic structure from hue data, which naturally lies on the unit circle. The probability density function of the Von Mises distribution is defined as:

$$
f(x \mid \mu, \kappa) = \frac{1}{2\pi I_0(\kappa)} \exp[\kappa \cos(x - \mu)],
$$

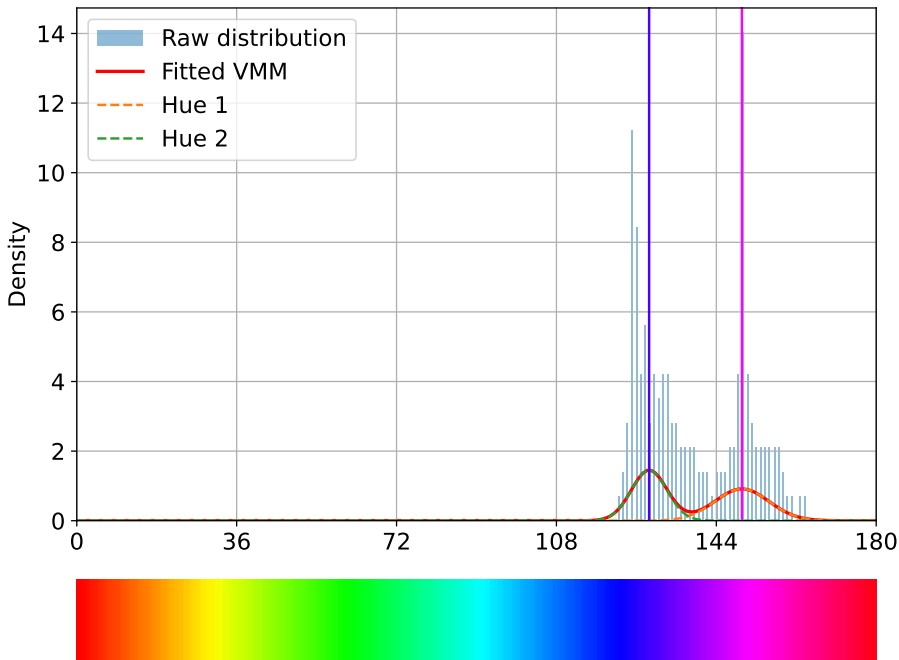

Figure 4: Visualization of a VMM Fitted to Hue Data: The method enables the decomposition of a mixed Hue distribution into several color components.

where $\mu$ is the mean direction, $\kappa$ is the concentration parameter, and $I_0$ denotes the modified Bessel function of the first kind of order zero.

A VMM models the hue distribution as a weighted sum of $K$ such components:

$$p(x) = \sum_{k=1}^{K} \pi_k f(x \mid \mu_k, \kappa_k),$$

where $\pi_k$ are the mixture weights satisfying $\sum_k \pi_k = 1$.

Given sampled values from the hue distribution as input, the model outputs the parameters of each Von Mises component. Parameter estimation is performed using the Expectation-Maximization (EM) algorithm, analogous to that used in Gaussian Mixture Models. In the absence of prior knowledge about the number of hue clusters, we evaluate a set of candidate models with varying component numbers and select the optimal one based on the Bayesian Information Criterion (BIC).

A sample frame from a real-world stage lighting sequence is visualized in Fig. 4. It successfully decomposes the mixed hue distribution into two distinct color components, making it a promising submodule for future studies.

## D  VISUALIZATION RESULTS

To provide an intuitive understanding of our model's performance on the stage lighting generation task, we visualize the output of the Skip-BART model, ground-truth and other two methods lighting control sequences, as shown in Fig. 5a and Fig. 5b. Each example consists of the following four parts:

- **Ground Truth (GT):** The ground-truth lighting sequence at each time frame. The color of each column directly represents the actual stage lighting color and brightness at that moment.
- **Skip-BART:** The lighting sequence predicted by the Skip-BART model.

- **Ablation Study:** The lighting sequence predicted by the BART model.

- **Rule Based:** The lighting sequence predicted by rule based method.

For Ground Truth, the color of each column directly represents the actual stage lighting color and brightness at that moment. For other three models, the color of each column reflects the model's prediction of lighting color and brightness.

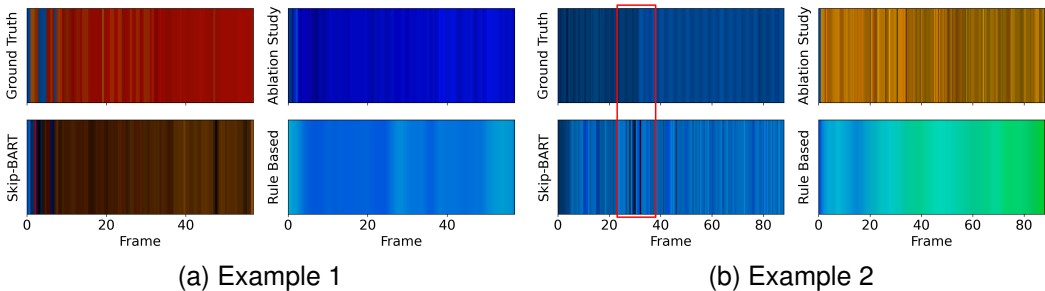

(a) Example 1    (b) Example 2

Figure 5: Visualization of lighting sequences generated by different methods. The top row shows the input Mel spectrogram, the middle row is the ground-truth lighting sequence, and the bottom row is the predicted sequence from the Skip-BART model. Each color represents a unique combination of lighting color and brightness over time. The red box in (b) highlights a representative segment where Skip-BART closely matches the temporal lighting structure of the ground truth.

In addition, we present a typical case in Fig. 5b, where the red box highlights a segment transition that Skip-BART successfully identifies, aligning well with the ground truth. Following this transition, both sequences exhibit a brighter light than before. Despite these strengths, Skip-BART still has certain limitations. It occasionally exhibits overly strong local fluctuations and fails to maintain the long-term rhythmic stability observed in the ground truth. This suggests that while the model handles short-term dynamics effectively, its global temporal structure modeling remains an area for improvement. The ablation variant, while visually distinct in color, still demonstrates some ability to follow the underlying temporal structure. In contrast, the rule-based method, relying solely on emotion mapping, tends to produce slow, smooth transitions, typically from blue to cyan to green, which results in outputs that lack the realism and complexity of human-designed lighting.

Overall, Skip-BART is built through transfer learning and pre-training so it learned latent music representations. Also, it is supervised trained, which means it naturally learned the lighting patterns present in the data. Similar to most generative models, it also incorporates controllable randomness through the temperature parameter. This allows the model to move beyond a single deterministic output and introduce variation that may appear some "creativity".

# E  HUMAN STUDY DETAILS

## E.1  STUDY SETUP

We recruited 38 respondents through social media platforms and live music venues, with the human evaluation spanning half a month. The participants included 20 males and 18 females aged 18–55 (predominantly 18–30), among whom 9 had professional experience in lighting design, music production, or stage art, and over half were frequent live music attendees. All participants reported normal hearing, normal or corrected-to-normal vision, and no history of color blindness or color weakness.

The questionnaire was structured as follows:

- **3 music pieces**: Each sampled randomly from the testing set, lasting 20 seconds.

- **4 objects per group**: Ground Truth, Skip-BART, Skip-BART (w/o skip connection) (i.e., Ablation Study), and Rule-based.

- **6 evaluation dimensions**: Emotional Match Between Lighting and Music, Visual Impact, Rhythmic Synchronization Accuracy, Smoothness of Lighting Transitions, Immersive Atmosphere Intensity, and Innovative Surprise (Erdmann et al., 2025).

Each music piece constituted one group, including four videos corresponding to the four objects, with the music serving as background sound and a dynamic color block representing the light changes. Within each music group, the four videos were presented in a randomized order to eliminate order effects. Participants were required to rate each video under each music group on the six dimensions using a 7-point Likert scale (1 = very dissatisfied, 7 = very satisfied).

### E.2    QUESTIONNAIRE CONTENT

## Livehouse Stage Lighting Experience Questionnaire

Dear Respondent: We are researchers conducting a study on Livehouse stage lighting experiences. Thank you for participating in this survey, which will take approximately 15 minutes to complete. Your responses will be kept strictly confidential and used solely for statistical analysis.

### PART 1: DEMOGRAPHICS

Q1. What is your age?
☐ Under 18    ☐ 18–24    ☐ 25–30
☐ 31–40    ☐ 41–50    ☐ 51 and above

Q2. What is your gender?
☐ Male    ☐ Female    ☐ Other

Q3. Current occupation: ______

Q4. Highest education level:
☐ High school    ☐ Junior college
☐ Bachelor's    ☐ Master's or above

Q5. Professional expertise (multiple):
☐ Lighting design    ☐ Music production
☐ Stage art    ☐ None

### PART 2: MUSIC BEHAVIORS

Q6. Frequently listened genres (multiple):
☐ Rock    ☐ Electronic    ☐ Hip-hop
☐ Pop    ☐ Metal    ☐ Jazz
☐ Classical    ☐ Other: ______

Q7. Livehouse attendance frequency:
☐ Weekly+    ☐ 2-3/month    ☐ Monthly
☐ Quarterly    ☐ Biannually-

Q8. Openness to new experiences:
1 (Very conservative) – 2 – 3 – 4 – 5 – 6 – 7 (Very proactive)

Q9. Lighting impact perception:
1 (Very little) – 2 – 3 – 4 – 5 – 6 – 7 (Very much)

### PART 3: LIGHTING EVALUATION

Instructions: Imagine you are at a Livehouse venue. Rate the following 3 groups of lighting effects based on your simulated experience. Each group contains 4 videos (20 seconds each) with the same background music but different lighting models.

- Q10-Q21.

  (Group X, Music Track Y) Please rate the Livehouse stage lighting effects based on the following 6 dimensions:

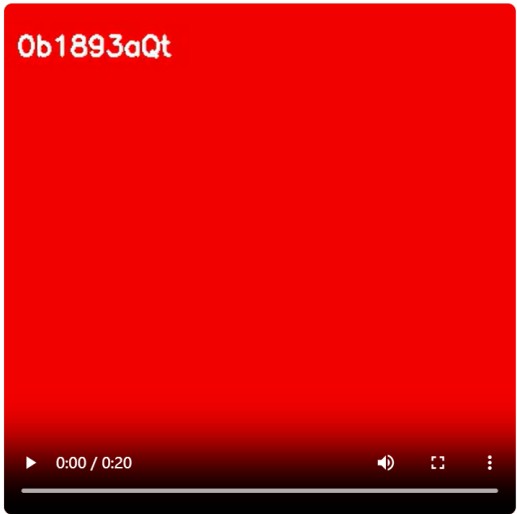

Figure 6: Video Interface Example: The upper-left labels serve as internal markers for video randomization to aid identification, with respondents unaware of their purpose.

| Evaluation Dimension | Rating Scale (1=Lowest, 7=Highest) | | | | | | |
|---|---|---|---|---|---|---|---|
| 1. Emotional music-light match | 1 | 2 | 3 | 4 | 5 | 6 | 7 |
| 2. Visual impact intensity | 1 | 2 | 3 | 4 | 5 | 6 | 7 |
| 3. Rhythmic synchronization | 1 | 2 | 3 | 4 | 5 | 6 | 7 |
| 4. Transition smoothness | 1 | 2 | 3 | 4 | 5 | 6 | 7 |
| 5. Immersive atmosphere | 1 | 2 | 3 | 4 | 5 | 6 | 7 |
| 6. Innovation surprise | 1 | 2 | 3 | 4 | 5 | 6 | 7 |

*Here $X \in \{1, 2, 3\}$ represents the three music groups, and $Y \in \{1, 2, 3, 4\}$ represents the four objects, including Ground Truth (GT), Skip-BART (Ours), Skip-BART (w/o skip connection), and the Rule-based method. For simplicity, we refer to Skip-BART (w/o skip connection) as Ablation Study in the appendix. To ensure fairness, within each group, the different models are presented in random orders for each audience. Additionally, the structural annotations (e.g. demographics, group divisions, and technical model names) served as internal organizational markers and were not displayed to respondents.*

**PART 4: CONCLUSION AND ACKNOWLEDGMENT**

Q22. Thank you for taking the time to participate in this survey! Your feedback is invaluable to us. If you wish, please leave your contact information (e.g., email, phone number) for potential follow-up communication:______

### E.3   DATA AGGREGATION

The composite variables were systematically constructed through mean aggregation of multidimensional scores across experimental conditions. For each lighting pattern's dimensional evaluation, final composite scores were calculated by averaging metric performances from three distinct background music groups. As exemplified in the main text, the notation (Ground Truth, Emotion) represents the composite variable for the Ground Truth lighting pattern's emotional match between the lighting and music dimensions, derived through mean aggregation of scores across all three background music conditions. Similarly, (Ground Truth, Overall) denotes the omnibus composite variable for Ground Truth, constructed by grand mean aggregation across all six evaluation dimensions and three musical conditions. This standardized mean-based composite variable construction methodology was uniformly applied to all experimental lighting patterns to ensure comparative consistency.

To refine our descriptive statistical analysis, the medians and plurality of the scores for the four objects across the six metrics are reported in Table 6.

Table 6: Median and Plurality Scores: The **bold** text represents the best result, while the underlined text indicates the second-best result. For the plurality, we retain the highest score when multiple pluralities exist.

| | Emotion | Impact | Rhythm | Smoothness | Atmosphere | Surprise | Overall |
|---|---|---|---|---|---|---|---|
| | | | | **Median** | | | |
| **Ground Truth** | 4.33 | **4.67** | **4.67** | **4.67** | **4.67** | **4.50** | **4.58** |
| **Skip-BART** | **4.67** | 4.50 | 4.50 | 4.33 | 4.33 | 4.17 | 4.42 |
| **Ablation Study** | 4.33 | 3.83 | **4.67** | 4.33 | 4.17 | 3.33 | 4.11 |
| **Rule-based** | 3.17 | 2.33 | 2.17 | 2.50 | 2.17 | 1.67 | 2.33 |
| | | | | **Plurality** | | | |
| **Ground Truth** | 4.33 | **5.33** | 4.67 | **5.00** | 5.00 | **5.00** | **5.17** |
| **Skip-BART** | **4.67** | 3.67 | **5.33** | 4.33 | **5.67** | 4.33 | 4.72 |
| **Ablation Study** | **4.67** | 4.00 | 4.67 | 4.33 | 5.00 | 4.67 | 4.61 |
| **Rule-based** | 4.00 | 1.00 | 1.00 | 1.00 | 1.00 | 1.00 | 1.00 |

### E.4 SIGNIFICANCE ANALYSIS

Given that the study involved repeated measurements of four methods across multiple metrics by the same group of subjects, a Repeated Measures ANOVA was selected for statistical analysis. This choice was made to effectively analyze the within-subject dependencies and to illustrate the advantages of the four methods across six metrics. To control for Type I errors that can arise from multiple comparisons, the Bonferroni correction method was employed to adjust the significance level.

The overall scores of the four methods were used as the dependent variable. The Mauchly sphericity test indicated that Mauchly's $W = 0.32$, $p < 0.001$, necessitating the application of the Greenhouse-Geisser correction. The results revealed a significant main effect of the four methods, $F(1.7, 63) = 61$, $p < 0.001$, $\eta^2 = 0.62$, suggesting significant differences in the overall scores among the four types of videos. *Here, $W$ denotes the Mauchly test statistic for sphericity, while $p$ signifies the probability value. The $F$ stands for the $F$ - ratio in ANOVA, with the figures in parentheses indicating the degrees of freedom (numerator, denominator), and $\eta^2$ serves as the measure of partial eta - squared effect size.* Subsequently, post hoc paired t-tests with Bonferroni correction were conducted.

The results, as shown in Table 5, demonstrated that the scores of the other three methods were significantly higher than those of the Rule-based method. In terms of specific pairwise comparisons, the score of Ground Truth was significantly higher than that of Ours (with or without skip connection) ($p = 0.003$). Although the Ground Truth score was higher than that of Skip-BART, and the Skip-BART score was higher than that of Ablation Study, these differences were not statistically significant.

Following the analysis of overall scores, a Repeated Measures ANOVA was also conducted on the scores of the four video types under six metrics. Similar to the previous procedure, post hoc paired t-tests with Bonferroni correction were performed. The pairwise comparison results are summarized in Table 7.

### E.5 CROSS DOMAIN EVALUATION

In this section, we provide the detailed statistical results of the cross-domain evaluation, as shown in Tables 8 and 9, which include three songs of different styles and evaluations from 30 users. Notably, our method consistently achieves higher scores compared to rule-based methods, demonstrating significant improvements.

Table 7: Pairwise Comparisons among Ground Truth (GT), Skip-BART (SB), Ablation Study (AS), and Rule-Based (RB): The symbols * and *** denote that the significance level (p) for the difference in means ($\Delta M$) is less than 0.05 or 0.001, respectively.

| Metrics | Comparison | $\Delta M$ | SD | p | Metrics | Comparison | $\Delta M$ | SD | p |
|---|---|---|---|---|---|---|---|---|---|
| **Emotion** | GT vs SB | -0.19 | 0.15 | 1.000 | **Impact** | GT vs SB | 0.09 | 0.13 | 1.000 |
| | GT vs AS | 0.19 | 0.11 | 0.598 | | GT vs AS | 0.70 | 0.15 | $< 0.001$*** |
| | GT vs RB | 1.38 | 0.25 | $< 0.001$*** | | GT vs RB | 1.83 | 0.23 | $< 0.001$*** |
| | SB vs AS | 0.39 | 0.14 | 0.045* | | SB vs AS | 0.61 | 0.14 | 0.001*** |
| | SB vs RB | 1.57 | 0.23 | $< 0.001$*** | | SB vs RB | 1.75 | 0.23 | $< 0.001$*** |
| | AS vs RB | 1.18 | 0.22 | $< 0.001$*** | | AS vs RB | 1.13 | 0.18 | $< 0.001$*** |
| **Rhythm** | GT vs SB | 0.11 | 0.12 | 1.000 | **Smoothness** | GT vs SB | 0.30 | 0.14 | 0.234 |
| | GT vs AS | 0.08 | 0.14 | 1.000 | | GT vs AS | 0.19 | 0.16 | 1.000 |
| | GT vs RB | 2.07 | 0.26 | $< 0.001$*** | | GT vs RB | 2.06 | 0.23 | $< 0.001$*** |
| | SB vs AS | -0.04 | 0.12 | 1.000 | | SB vs AS | -0.11 | 0.16 | 1.000 |
| | SB vs RB | 1.96 | 0.21 | $< 0.001$*** | | SB vs RB | 1.76 | 0.23 | $< 0.001$*** |
| | AS vs RB | 1.99 | 0.25 | $< 0.001$*** | | AS vs RB | 1.87 | 0.20 | $< 0.001$*** |
| **Atmosphere** | GT vs SB | 0.17 | 0.14 | 1.000 | **Surprise** | GT vs SB | 0.51 | 0.16 | 0.014* |
| | GT vs AS | 0.38 | 0.15 | 0.088 | | GT vs AS | 0.84 | 0.16 | $< 0.001$*** |
| | GT vs RB | 1.72 | 0.25 | $< 0.001$*** | | GT vs RB | 1.99 | 0.23 | $< 0.001$*** |
| | SB vs AS | 0.21 | 0.13 | 0.619 | | SB vs AS | 0.33 | 0.17 | 0.318 |
| | SB vs RB | 1.55 | 0.21 | $< 0.001$*** | | SB vs RB | 1.48 | 0.21 | $< 0.001$*** |
| | AS vs RB | 1.34 | 0.18 | $< 0.001$*** | | AS vs RB | 1.15 | 0.17 | $< 0.001$*** |

Table 8: Median and Plurality Scores of Cross-Domain Evaluation

| | Emotion | Impact | Rhythm | Smoothness | Atmosphere | Surprise | Overall |
|---|---|---|---|---|---|---|---|
| | | | | **Median** | | | |
| **Skip-BART** | **4.25** | **4.11** | **4.80** | **4.72** | **4.52** | **4.07** | **4.30** |
| **Ablation Study** | 3.73 | 3.89 | 4.59 | 4.58 | 4.00 | 3.71 | 4.13 |
| **Rule-based** | 3.50 | 2.83 | 3.05 | 3.89 | 3.14 | 2.72 | 3.08 |
| | | | | **Plurality** | | | |
| **Skip-BART** | **4.00** | **3.00** | 3.33 | **5.67** | **4.67** | **4.33** | **3.94** |
| **Ablation Study** | 3.33 | **3.00** | **4.67** | 4.00 | **4.00** | 3.33 | 3.61 |
| **Rule-based** | 3.33 | **3.00** | 3.00 | 4.67 | 3.00 | 3.00 | 2.56 |

Table 9: Pairwise Comparisons of Cross-Domain Evaluation

| Metrics | Comparison | $\Delta M$ | SD | p | Metrics | Comparison | $\Delta M$ | SD | p |
|---|---|---|---|---|---|---|---|---|---|
| **Emotion** | SB vs AS | 0.37 | 0.20 | 0.249 | **Impact** | SB vs AS | 0.24 | 0.17 | 0.45 |
| | SB vs RB | 0.70 | 0.21 | 0.007** | | SB vs RB | 1.13 | 0.23 | $< 0.001$*** |
| | AS vs RB | 0.33 | 0.22 | 0.435 | | AS vs RB | 0.89 | 0.18 | $< 0.001$*** |
| **Rhythm** | SB vs AS | 0.16 | 0.16 | 1.00 | **Smoothness** | SB vs AS | 0.19 | 0.10 | 0.21 |
| | SB vs RB | 1.39 | 0.25 | $< 0.001$*** | | SB vs RB | 0.73 | 0.21 | 0.004** |
| | AS vs RB | 1.23 | 0.21 | $< 0.001$*** | | AS vs RB | 0.54 | 0.21 | 0.047* |
| **Atmosphere** | SB vs AS | 0.37 | 0.17 | 0.12 | **Surprise** | SB vs AS | 0.12 | 0.14 | 1.00 |
| | SB vs RB | 1.07 | 0.21 | $< 0.001$*** | | SB vs RB | 0.96 | 0.23 | $< 0.001$*** |
| | AS vs RB | 0.70 | 0.17 | $< 0.001$*** | | AS vs RB | 0.83 | 0.22 | 0.002** |

