# OpenReview forum: "Automatic Stage Lighting Control: Is it a Rule-Driven Process or Generative Task?"
_ICLR.cc/2026/Conference — ICLR 2026 Poster_

### Official Review · Reviewer_JUmG · 2025-10-19

**Soundness:** 3
**Presentation:** 4
**Contribution:** 4
**Rating:** 8
**Confidence:** 4

**Summary:**

This paper addresses the task of Automatic Stage Lighting Control (ASLC) by framing it as an end-to-end generative problem for the first time. The authors propose a novel end-to-end ASLC model, Skip-BART, and introduce a dedicated stage lighting dataset.

**Strengths:**

- The paper addresses an important task of the stage lighting generation.
- The paper's contribution is significant by introducing the end-to-end ASLC task.
- The paper is well written and easy to follow.
- Thorough experiments showing the effectiveness of the proposed method.

**Weaknesses:**

- The description of the pre-training stage lacks sufficient detail (see the Questions section).
- Some claims are not adequately supported by experimental evidence:
    - Allowing the model to identify which tokens have been replaced by [MASK] assists the learning process.
    - The role and impact of incorporating the discriminator
- The method does not support real-time lighting generation. While not a critical limitation, real-time generation is a compelling use case for ASLC, particularly in live performances. The reliance on an encoder-decoder architecture restricts the model to offline processing, preventing it from handling streaming music input.

**Questions:**

- In the pre-training stage, is it the decoder or the regressor that is responsible for recovering the embeddings of the masked segments?
- What are the targets of the decoder and the regressor during pre-training?
- What is the architectural design of the regressor?

---

> ### Author Response · Authors · 2025-11-20
> **Response to Reviewer JUmG (Part 1)**
>
> We are grateful for your thoughtful feedback. Your assessment is valuable for us to improve our work. We are glad to reply to your questions one by one.
>
> > W1: The description of the pre-training stage lacks sufficient detail (see the Questions section).
>
> Thank you for your insightful comment. We have added a new section in the Appendix A to provide further details on the pre-training stage for reproducibility. Please also refer to our responses to `Q1` to `Q3` below for more details.
>
>
>
> > W2: Some claims are not adequately supported by experimental evidence:
> > - Allowing the model to identify which tokens have been replaced by [MASK] assists the learning process.
> > - The role and impact of incorporating the discriminator
>
> Thank you forpointing this out. First, we'd like to clarify that these claims derive from a previous work[1] and are not originally introduced in this paper. To provide further evidence, we conducted an extended ablation study using quantitative metrics. The results show that without the two mechanisms, performance decreases across all metrics. The detailed experiment results are presented below and have been added to Table 1 in the manuscript.
>
> |                             | Hue RMSE ($\downarrow$) | Value RMSE ($\downarrow$) | Hue MAE ($\downarrow$) | Value MAE ($\downarrow$) | Hue $\mathrm{corr}(\|\Delta\|) (\times10^{-2})$ ($\uparrow$) | Value $\mathrm{corr}(\|\Delta\|) (\times10^{-2})$ ($\uparrow$) |
> | --------------------------- | ----------------------- | ------------------------- | ---------------------- | ------------------------ | --------------------- | --------------------- |
> | Ours                        | **36.13**               | **60.74**                 | **28.72**              | **51.27**                | **0.88**              | **2.94**              |
> | pre-train w/o discriminator | 50.40                   | 68.09                     | 41.52                  | 56.54                    | 0.46                  | 1.13                  |
> | pre-train w/o random [MASK] | 49.97                   | 64.45                     | 42.07                  | 52.63                    | 0.54                  | 1.11                  |
>
>
>
> [1] Zhao, Zijian, et al. "Mining limited data sufficiently: A bert-inspired approach for csi time series application in wireless communication and sensing." arXiv preprint arXiv:2412.06861 (2024).

---

> ### Author Response · Authors · 2025-11-20
> **Response to Reviewer JUmG (Part 2)**
>
> > W3: The method does not support real-time lighting generation. While not a critical limitation, real-time generation is a compelling use case for ASLC, particularly in live performances. The reliance on an encoder-decoder architecture restricts the model to offline processing, preventing it from handling streaming music input.
>
> Thank you for your insightful comment. We acknowledge that a shortcoming of our method is the lack of support for real-time control, which is also a limitation shared by most previous methods and constitutes a significant research question for future work.
>
> Indeed, the Automatic Stage Lighting Control (ASLC) is still in its early stages. Most existing works [1-3], including ours, focus on offline control of a single primary light, primarily aiming at assisting and inspiring human lighting engineers. We agree that real-time and multi-light control is crucial for production-grade systems. However, it remains a widespread and significant challenge for the entire field because of several realistic constraints:
>
> **(i) Light Configuration Heterogeneity Among Stages:** For ASLC to be applied in real-world setting, detailed light configurations must be taken into account. Unfortunately, the amount and positioning of lights often differ among stages, meaning that the same control input can produce different visual performances. To tackle this, light configuration must be incorporated as part of the input, or a pre-training with fine-tuning paradigm can be developed, allowing the pre-trained model to be quickly fine-tuned according to different stage setups.
>
> **(ii) High Dynamics of Songs:** A real-time ASLC framework must have the capacity to anticipate future changes in a song, as lighting adjustments must be executed within seconds. However, predicting a song accurately is challenging due to its artistic nature, where surprises are the charm of music. Nevertheless, we agree that enabling real-time and online lighting generation would greatly simplify the overall workflow and significantly enhance the usability of human–AI interaction. We will continue to explore this direction in future work.
>
> Additionally, we want to argue that for most real-world livehouse applications, an offline light generation solution is sufficient, becauseMost live performances require the Program (PGM) to be prepared in advance. During the performance, singers and musicians typically follow a click track in their In-Ear Monitors to ensure alignment with the PGM. This indicates that the entirety of the music can be known well in advance of the performance, allowing for pre-generated and prepared lighting that aligns with the PGM.
>
> We believe our work makes sufficient contribution to the field by introducing a novel perspective by framing ASLC as a creative generation task. Furthermore, we are releasing the first ASLC dataset that supports multiple light extractions, which will serve as a valuable resource for addressing these practical challenges in future research.  Additionally, in the Appendix, we provide a multiple hue extraction method that efficiently extracts different components from each frame based on the Von Mises Mixture Model, offering an effective mechanism for ground truth extraction in future work.
>
> In response to your suggestion, we have revised the limitations and future work sections of our manuscript. We hope that these clarifications and adjustments sufficiently address your concerns.
>
> [1] Hsiao, Shih-Wen, Shih-Kai Chen, and Chu-Hsuan Lee. "Methodology for stage lighting control based on music emotions." Information sciences 412 (2017): 14-35.
>
> [2] Hu, Jiayun, et al. "Music2Palette: Emotion-aligned Color Palette Generation via Cross-Modal Representation Learning." Proceedings of the 33rd ACM International Conference on Multimedia. 2025.
>
> [3] Moon, Chang Bae, et al. "Mood lighting system reflecting music mood." Color Research & Application 40.2 (2015): 201-212.

---

> > ### Comment · Reviewer_JUmG · 2025-11-20
> >
> > Thank you for the detailed explanations. I believe the paper has made sufficient contributions, and my assessment is already quite positive. The reason I bring up the real-time generation problem is that there are scenarios where musicians jam on stage (meaning the music is not known in advance). I see this as a very interesting application that requires real-time lighting generation and highly encourages the authors to explore it in future works.

---

> > > ### Author Response · Authors · 2025-11-20
> > > **Thank you for Reviewer's reply**
> > >
> > > Dear Reviewer JUmG,
> > >
> > > Thank you once again for your thorough review and constructive feedback regarding our submission. We also agree that real-time light control could provide intriguing and surprising benefits for lives and jams, and we aim to explore this in our future work.
> > >
> > > ICLR 2026 Submission 4854 Author

---

> ### Author Response · Authors · 2025-11-20
> **Response to Reviewer JUmG (Part 3)**
>
> > Q1: In the pre-training stage, is it the decoder or the regressor that is responsible for recovering the embeddings of the masked segments?
>
> Thank you for your insightful question. You are correct -- both the decoder and the regressor are responsible for recovering the original input sequence from the encoder.
>
> During pre-training, we follow a similar setting as the MLM task used in the original BART model [1]. Specifically, in each epoch, for a given embedding sequence $e = \{e_1, e_2, \ldots, e_n\}$, we first randomly mask some of the embeddings to generate the input to the encoder, denoted as $\tilde{e} = \{\tilde{e}_1, \tilde{e}_2, \ldots, \tilde{e}_n\}$. If $\tilde{e}_i$ is chosen, it becomes the random [MASK] token.
>
> Otherwise, it retains its original value $e_i$. For the decoder, we utilize the shift-right ground truth as input, i.e., $\{[\text{SOS}], e_1, e_2, \ldots, e_{n-1}\}$, and it will output the hidden features $f = \{f_1, f_2, \ldots, f_n\}$. These features are then fed into a regressor (a simple MLP) to generate the estimated input sequence, $\hat{e} = \{\hat{e}_1, \hat{e}_2, \ldots, \hat{e}_n\}$.
>
> [1] Lewis M, Liu Y, Goyal N, et al. BART: Denoising sequence-to-sequence pre-training for natural language generation, translation, and comprehension. Proceedings of the 58th Annual Meeting of the Association for Computational Linguistics, 2020: 7871-7880.
>
> > Q2: What are the targets of the decoder and the regressor during pre-training?
>
> Thank you for your insightful question. The target of the decoder and regressor is to accurately recover the masked input sequence (i.e., music embedding).
>
> The loss function is defined as shown in Eq. (4). By this approach, we hope the BART backbone can learn the fundamental structure of music sequences, providing a solid starting point for the downstream light generation task. Although the input to the decoder differs between pre-training (music) and fine-tuning (light), it should still be beneficial according to the theory of cross-modality transfer learning [1], which is also supported by our ablation study.
>
> [1] Niu S, Liu Y, Wang J, et al. A decade survey of transfer learning (2010–2020). IEEE Transactions on Artificial Intelligence, 2021, 1(2): 151-166
>
> > Q3: What is the architectural design of the regressor?
>
> Thank you for your insightful question. In our experiments, we utilize a three-layer MLP with a LeakyReLU activation function to map the output of the decoder $f$ to the recovered (estimated) original input sequence $e$.
>
> ---
> Finally, we appreciate your recognition of our work and your constructive suggestions. Incorporating these technical details has indeed made the paper clearer, more thorough in its theoretical presentation, and more reproducible.

---

### Official Review · Reviewer_PdJ6 · 2025-10-31

**Soundness:** 3
**Presentation:** 3
**Contribution:** 3
**Rating:** 8
**Confidence:** 4

**Summary:**

This paper presents an engaging and well-articulated exploration of automatic stage lighting control, framing the problem not as a fixed, rule-driven process but as a generative task. The authors propose a deep learning model, Skip-BART, that translates audio into light parameters and seeks to bridge the gap between traditional handcrafted lighting rules and creative, data-driven systems. Alongside the model, the paper introduces a dataset and outlines a methodology that combines pre-training, fine-tuning, and sampling procedures.

**Strengths:**

I find the work’s concept compelling. The authors do a good job explaining why rule-based and procedural systems struggle to capture the expressive and performative nature of lighting design. Framing lighting generation as a creative modeling problem rather than a deterministic optimization task opens promising ground between creative AI and live performance technology.

The interdisciplinary scope of the paper feels well balanced - it takes technical rigor seriously without losing sight of the artistic intent behind lighting. The proposed model and dataset are well described, which makes the work easy to follow and potentially reproducible. I also appreciate that the evaluation includes both quantitative metrics and human judgment, acknowledging that creative systems can’t be fully understood through quantitative metrics alone.

The results indicate that the model performs comparably to professional lighting engineers within the tested scenarios, providing convincing support for the paper’s overall argument.

**Weaknesses:**

Overall, the paper is strong, but I think the framing of its contribution could be clearer. The discussion of data limitations in prior work is well justified - the authors point out that existing methods rely on small, coarse-grained, or biased datasets, which constrain performance. However, this feels more like a practical limitation than a fundamental research gap. It would help if the paper made clearer whether its main contribution is technical (a new model that performs better) or conceptual (reframing stage lighting as a generative, creative process). It currently aims to do both, but the balance between these two dimensions isn’t always explicit.

The human evaluation adds a relevant perspective, and it’s great that the authors use both quantitative and perceptual measures. That said, readers would benefit from a brief summary of main participant details and evaluation criteria in the main text (the info is in Appendix D, but it would be better to have a glimpse of it earlier). The main section does report the sample size (n=38), number of pieces, and metrics used, but a concise overview of demographics and procedure would make the “human-level performance” claim easier to interpret at a glance.

Lastly, while the paper acknowledges some constraints in passing - such as occasional rhythm instability and the need for more long-term temporal coherence - these points are scattered across the main text and appendices. A short, dedicated Limitations and Future Work section would help consolidate them, making it clear where the approach currently struggles (e.g., dataset bias, dependence on specific musical genres, or limited cross-domain evaluation beyond music-driven contexts). At moments, the discussion leans heavily toward optimization details like sampling temperature and adaptive weighting. While technically relevant, this can “dilute” the creative framing that makes the generative perspective so compelling in the first place.


Overall, I would recommend this work for acceptance once the points raised are addressed. The paper makes a meaningful contribution to the emerging conversation on generative modeling for creative control systems, presenting both a technical framework and an insightful reframing of stage lighting as a creative, data-driven process. Skip-BART is well-motivated and empirically supported and consistently surpasses rule-based baselines. With a few refinements (especially in clarifying the balance between the technical and conceptual contributions, summarizing key details of the human evaluation in the main text, and consolidating the discussion of limitations) I believe this work could serve as a strong reference point for future research on AI-assisted performance design and the broader intersection of generative models and creative practice.

**Questions:**

- Could the authors elaborate on whether Skip-BART is capable of generating new or unexpected lighting behaviors, or whether it primarily reproduces stylistic patterns learned from the training data? The paper strongly supports the generative framing, but most evidence focuses on similarity to ground truth and outperforming rule-based methods. Clarifying how “generative” this process truly is (e.g. in terms of diversity or novelty) would strengthen the conceptual contribution.

- The human study is clearly designed and statistically analyzed, with 38 participants evaluating lighting across six perceptual dimensions. Still, a brief summary of participant demographics and evaluation setup in the main text (currently detailed in Appendix D) would make the study easier to follow and lend more transparency to the “human-level performance” claim.

- ​The cross-domain evaluation (using Suno-generated folk, R&B, and jazz samples) is a nice addition, but it could be discussed more explicitly in terms of limitations. For instance, how robust is the model to genres or performance settings that differ substantially from those represented in the RPMC-L2 dataset? This reflection would clarify where the method generalizes well and where domain adaptation might be needed (paving the way and presenting a gap for further work).

- ​While some constraints are mentioned in the discussion and appendices (e.g., rhythm
stability and dataset scope), consolidating them in a concise section would help readers understand the boundaries of the approach. This could include points such as dataset bias toward rock/metal performances, limited exploration of multi-light setups, and future directions for broader aesthetic or cross-genre applications.

---

> ### Author Response · Authors · 2025-11-20
> **Response to Reviewer PdJ6 (Part 1)**
>
> Thank you for your thoughtful feedback. Your feedback is valuable for us to improve our work. We are glad to reply to your concerns and questions one by one.
>
> > W1: Overall, the paper is strong, but I think the framing of its contribution could be clearer. The discussion of data limitations in prior work is well justified - the authors point out that existing methods rely on small, coarse-grained, or biased datasets, which constrain performance. However, this feels more like a practical limitation than a fundamental research gap. It would help if the paper made clearer whether its main contribution is technical (a new model that performs better) or conceptual (reframing stage lighting as a generative, creative process). It currently aims to do both, but the balance between these two dimensions isn’t always explicit.
>
> Thank you for your insightful comment. First, regarding the contributions of this paper, while we would like to claim both both conceptual and technical contributions, our primary emphasis is on the conceptual innovation. From a conceptual standpoint, our work provides a novel perspective for ASLC by viewing it as a creative generation task, which fundamentally differs from the rule-based mapping paradigms of previous methods. The technical aspects of our work are designed to address this generation task effectively. Through our network structure and training process design, our method successfully achieves similar-to-human-level performance, establishing a solid baseline for future research in the proposed generation-based ASLC task. To clarify this further, we have revised the contribution summary at the end of the Introduction section in accordance with your suggestion.
>
> Second, we believe that data limitations are only one of the issues faced by most prior work. Additionally, many previous approaches rely on rule-based methods that  first classify music pieces into different types based on dimensions like style, chord, and emotion, then map each category to a predefined light pattern. However, the relationship among these categories and stage lighting remains an open question, as the connections among them are not sufficiently clear [1]. Furthermore, the coarse-grained and low-accuracy classification exacerbates these issues due to dataset limitations. In contrast, our end-to-end method does not face the same rationality problems, as our training process already ensures that the model can learn effectively from limited datasets.
>
> [1] McDonald J, Canazza S, Chmiel A, et al. Illuminating music: Impact of color hue for background lighting on emotional arousal in piano performance videos[J]. Frontiers in Psychology, 2022, 13: 828699.
>
> > W2: The human evaluation adds a relevant perspective, and it’s great that the authors use both quantitative and perceptual measures. That said, readers would benefit from a brief summary of main participant details and evaluation criteria in the main text (the info is in Appendix D, but it would be better to have a glimpse of it earlier). The main section does report the sample size (n=38), number of pieces, and metrics used, but a concise overview of demographics and procedure would make the “human-level performance” claim easier to interpret at a glance.
>
> Thank you for insightful comment. We have added a brief summary at the beginning of Section 4.3 according to your suggestion.

---

> ### Author Response · Authors · 2025-11-20
> **Response to Reviewer PdJ6 (Part 2)**
>
> > W3: Lastly, while the paper acknowledges some constraints in passing - such as occasional rhythm instability and the need for more long-term temporal coherence - these points are scattered across the main text and appendices. A short, dedicated Limitations and Future Work section would help consolidate them, making it clear where the approach currently struggles (e.g., dataset bias, dependence on specific musical genres, or limited cross-domain evaluation beyond music-driven contexts). At moments, the discussion leans heavily toward optimization details like sampling temperature and adaptive weighting. While technically relevant, this can “dilute” the creative framing that makes the generative perspective so compelling in the first place.
>
> Thank you for your insightful comment. We have added the Limitation and Future Work section in the main text according to your suggestion. Specifically, (1) we mention the occasional rhythm instability and the need for stronger long-term temporal coherence in the Limitations and Future Work-Alternative Generation Technology Exploration part. We believe that emerging techniques may offer promising directions for improving model capability. (2) We mention limited exploration of multi-light setups in the Limitations and Future Work-Towards More Practical Scenarios part. Future research could investigate real-time and multi-light control, and we also outline several challenges that would need to be addressed. (3) We mention dataset bias and cross-domain limitations in the Limitations and Future Work-Generalizability Across Music Genres part. More data can usually lead to more robust and higher-quality results, but data in related domains are often scarce. We believe domain alignment methods offer a promising way to address this issue. (4) In addition, we discuss the use of richer modalities in the Limitations and Future Work-Richer Modality Incorporation part. Additional modalities may provide more informative signals.
>
> > W4: Overall, I would recommend this work for acceptance once the points raised are addressed. The paper makes a meaningful contribution to the emerging conversation on generative modeling for creative control systems, presenting both a technical framework and an insightful reframing of stage lighting as a creative, data-driven process. Skip-BART is well-motivated and empirically supported and consistently surpasses rule-based baselines. With a few refinements (especially in clarifying the balance between the technical and conceptual contributions, summarizing key details of the human evaluation in the main text, and consolidating the discussion of limitations) I believe this work could serve as a strong reference point for future research on AI-assisted performance design and the broader intersection of generative models and creative practice.
>
> We sincerely appreciate your positive assessment of our work. We hope this research can make a meaningful contribution to the advancement of AI-assisted performance design and the broader intersection of generative models and creative practice. We also hope that our responses have fully addressed your concerns.
>
>
> > Q1: Could the authors elaborate on whether Skip-BART is capable of generating new or unexpected lighting behaviors, or whether it primarily reproduces stylistic patterns learned from the training data? The paper strongly supports the generative framing, but most evidence focuses on similarity to ground truth and outperforming rule-based methods. Clarifying how “generative” this process truly is (e.g. in terms of diversity or novelty) would strengthen the conceptual contribution.
>
> Thank you for your insightful question. This is indeed  an interesting point. Similar to "Does current AIGC generate new content, or just imitate the human behavior in training set?", which we believe this remains an open, and perhaps even philosophical, question worthy of ongoing discussion. Skip-BART is built through transfer learning and pre-training so it learned latent music representations. Also, it is supervised trained, which means it naturally learned the lighting patterns present in the data. Similar to most generative models, it also incorporates controllable randomness through the temperature parameter. This allows the model to move beyond a single deterministic output and introduce variation that may appear some "creativity".
> Overall, we believe that current generative technologies, including Skip-BART, still operate largely within the paradigm defined by the training data. Even so, the stochastic component in the generation process can lead to outputs that differ from seen examples and display a degree of creative behavior.

---

> ### Author Response · Authors · 2025-11-20
> **Response to Reviewer PdJ6 (Part 3)**
>
> > Q2: The human study is clearly designed and statistically analyzed, with 38 participants evaluating lighting across six perceptual dimensions. Still, a brief summary of participant demographics and evaluation setup in the main text (currently detailed in Appendix D) would make the study easier to follow and lend more transparency to the “human-level performance” claim.
>
> Thank you for your insightful comment. We have added a brief summary at the beginning of Section 4.3 in accordance with your suggestion.
>
>
> > Q3: The cross-domain evaluation (using Suno-generated folk, R&B, and jazz samples) is a nice addition, but it could be discussed more explicitly in terms of limitations. For instance, how robust is the model to genres or performance settings that differ substantially from those represented in the RPMC-L2 dataset? This reflection would clarify where the method generalizes well and where domain adaptation might be needed (paving the way and presenting a gap for further work).
>
>
>
> Thank you for your insightful comment. First, we believe that our method demonstrates high robustness in cross-domain scenarios, as it achieves a similar level of human evaluation scores. This suggests that the model is learning a useful representation of music-to-light relationships and generalizing successfully to out-of-distribution data. However, according to deep learning theory, such a shift in data distribution inevitably leads to a performance decrease, even if the results appear acceptable based on our findings. In practice, users can retrain or fine-tune our method using their own datasets with specific scopes. Considering the data scarcity and collection difficulties in ASLC, unsupervised domain alignment presents a potential paradigm, as it requires only unlabeled data (i.e., music without corresponding light) in the target domain. We expect future research to investigate the potential improvements that such unsupervised domain alignment approaches can bring to our proposed method. We have added this discussion to the Limitations and Future Work section.
>
>
>
> > Q4: While some constraints are mentioned in the discussion and appendices (e.g., rhythm stability and dataset scope), consolidating them in a concise section would help readers understand the boundaries of the approach. This could include points such as dataset bias toward rock/metal performances, limited exploration of multi-light setups, and future directions for broader aesthetic or cross-genre applications.
>
>
> Thank you for your insightful suggestion. We have reorganized the content in the Limitations and Future Work section. Please refer to the details in our response to `W3`.
>
> ---
> Finally, we express our heartfelt gratitude for your detailed suggestions. We sincerely appreciate your recognition of our work and your constructive advice on improving the clarity of our manuscript.

---

### Official Review · Reviewer_1ux2 · 2025-11-02

**Soundness:** 3
**Presentation:** 3
**Contribution:** 4
**Rating:** 6
**Confidence:** 3

**Summary:**

This paper proposes Skip-BART, the first end-to-end deep learning framework for Automatic Stage Lighting Control (ASLC), reframing the task from a rule-driven classification problem into a generative modelling problem.
The authors adapt BART to generate lighting hue and intensity directly from audio input, introducing a skip-connection mechanism to align music and light on a frame-wise level. They also introduce transfer learning (leveraging PianoBART), pre-training via a masked language modelling objective, and a novel Restricted Stochastic Temperature-Controlled (RSTC) sampling for inference.
A new dataset, RPMC-L2, containing 699 stage lighting samples from live performances, is constructed for training and evaluation. Quantitative metrics (RMSE, MAE, corr(|∆|)) and a human evaluation with 38 participants demonstrate that Skip-BART significantly outperforms rule-based baselines and achieves performance statistically indistinguishable from professional lighting engineers

**Strengths:**

• The introduction of Skip-BART with skip connections for frame-level alignment is technically sound and novel.

• The method is well-motivated and supported by solid engineering: effective dataset construction, comprehensive ablations, and careful pre-training/fine-tuning strategies.

• The paper is clearly structured, with diagrams and methodological details that enhance understanding.

• This work opens a new research direction in artistic multimodal generation, bridging music information retrieval and stage performance automation.

**Weaknesses:**

• The dataset is domain-specific (rock/punk/metal), limiting generalization to genres such as pop, jazz, or classical. Cross-domain evaluation shows promise but remains narrow in scope.

• The pre-training details (e.g., MLM masking ratios, discriminator architecture) could be clarified further for full reproducibility.

• Real-time or multi-light control is not addressed, which would be crucial for production-grade systems.

**Questions:**

See Weakness

---

> ### Author Response · Authors · 2025-11-20
> **Response to Rviewer 1ux2 (Part 1)**
>
> Thank you for your thoughtful feedback. Your feedback is valuable for us to improve our work. We are glad to reply to your concerns and questions one by one.
>
> > W1: The dataset is domain-specific (rock/punk/metal), limiting generalization to genres such as pop, jazz, or classical. Cross-domain evaluation shows promise but remains narrow in scope.
>
> Thank you for your insightful comment. We acknowledge that the narrow data scope is a limitation of this work. We currently focus on rock, punk, and metal performances, which are dominant music genres played in livehouses with strong lighting effects. This setting provides a challenging but practically relevant testbed, where model must learn complex lighting. Besides, our experiments show positive results when evaluated on out-of-domain data. This suggests that the model is learning a useful representation of music-to-light relationships that is generalizable to other genres.
>
> Additionally, we would like to clarify that the main contribution of this paper is  a novel paradigm for Automatic Stage Lighting Control (ASLC), complemented by a framework that includes network structure and training methodologies. In practice, the proposed method is independent of the dataset, allowing users to train their individual models using their own datasets with specific scopes.
>
> In response to your comment, we have included these points in Section 5 Limitations and Future Work section of our updated manuscript.
>
> > W2: The pre-training details (e.g., MLM masking ratios, discriminator architecture) could be clarified further for full reproducibility.
>
> Thank you for your insightful suggestion. For the masking ratios in MLM, we set it as a random number sampling from the uniform distribution $U(0.15, 0.30)$ in each epoch. For the discriminator, we utilize the sequential classifier used in MidiBERT [1], which is a pioneering work in symbolic music classification field. Specifically, for the embedding sequence $e \in \mathbb{R}^{n,h}$ where $n$ is sequence length and $h$ represents feature dimension, the discriminator process it as:
>
> $$
> M_{attn}  = \text{Softmax}(\text{LinearLayer}(e)) \in \mathbb{R}_+^{n \times d}
> $$
>
> $$
> z  = \text{Flatten}(M_{attn}^T \cdot e) \in \mathbb{R}^{h d}
> $$
>
> $$
> y  = \text{Softmax}(\text{LinearLayer}(z)) \in \mathbb{R}_+^{2}
> $$
>
> where $M_{attn}$ is an attention matrix, $d$ is hidden dimension, and $y$ is a binary classification result. We have added these setup details in the Appendix A for reproducibility with full code and parameters open-sourced.
>
> [1] Chou, Yi-Hui, et al. "Midibert-piano: Large-scale pre-training for symbolic music understanding." arXiv preprint arXiv:2107.05223 2 (2021).

---

> ### Author Response · Authors · 2025-11-20
> **Response to Rviewer 1ux2 (Part 2)**
>
> > W3: Real-time or multi-light control is not addressed, which would be crucial for production-grade systems.
>
> Thank you for your insightful comment. Indeed, the Automatic Stage Lighting Control (ASLC) is still in its early stages. Most existing works [1-3], including ours, focus on offline control of a single primary light, primarily aiming at assisting and inspiring human lighting engineers. We agree that real-time and multi-light control is crucial for production-grade systems. However, it remains a widespread and significant challenge for the entire field because of several realistic constraints:
>
> **(i) Light Configuration Heterogeneity Among Stages:** For ASLC to be applied in real-world setting, detailed light configurations must be taken into account. Unfortunately, the amount and positioning of lights often differ among stages, meaning that the same control input can produce different visual performances. To tackle this, light configuration must be incorporated as part of the input, or a pre-training with fine-tuning paradigm can be developed, allowing the pre-trained model to be quickly fine-tuned according to different stage setups.
>
> **(ii) High Dynamics of Songs:** A real-time ASLC framework must have the capacity to anticipate future changes in a song, as lighting adjustments must be executed within seconds. However, predicting a song accurately is challenging due to its artistic nature, where surprises are the charm of music. Nevertheless, we agree that enabling real-time and online lighting generation would greatly simplify the overall workflow and significantly enhance the usability of human–AI interaction. We will continue to explore this direction in future work.
>
> Additionally, we want to argue that for most real-world livehouse applications, an offline light generation solution is sufficient, becauseMost live performances require the Program (PGM) to be prepared in advance. During the performance, singers and musicians typically follow a click track in their In-Ear Monitors to ensure alignment with the PGM. This indicates that the entirety of the music can be known well in advance of the performance, allowing for pre-generated and prepared lighting that aligns with the PGM.
>
> We believe our work makes sufficient contribution to the field by introducing a novel perspective by framing ASLC as a creative generation task. Furthermore, we are releasing the first ASLC dataset that supports multiple light extractions, which will serve as a valuable resource for addressing these practical challenges in future research.  Additionally, in the Appendix, we provide a multiple hue extraction method that efficiently extracts different components from each frame based on the Von Mises Mixture Model, offering an effective mechanism for ground truth extraction in future work.
>
> In response to your suggestion, we have revised the limitations and future work sections of our manuscript. We hope that these clarifications and adjustments sufficiently address your concerns.
>
> [1] Hsiao, Shih-Wen, Shih-Kai Chen, and Chu-Hsuan Lee. "Methodology for stage lighting control based on music emotions." Information sciences 412 (2017): 14-35.
>
> [2] Hu, Jiayun, et al. "Music2Palette: Emotion-aligned Color Palette Generation via Cross-Modal Representation Learning." Proceedings of the 33rd ACM International Conference on Multimedia. 2025.
>
> [3] Moon, Chang Bae, et al. "Mood lighting system reflecting music mood." Color Research & Application 40.2 (2015): 201-212.
>
> ---
> Finally, we express our heartfelt gratitude for your insightful questions and suggestions. They have been invaluable to our work. We hope our efforts will adequately address your concerns.

---

### Author Response · Authors · 2025-11-29
**Rebuttal Summary**

Dear Reviewers, ACs, and PCs,

We would like to thank the ACs and PCs for handling our paper, and we extend our gratitude to the reviewers for their expertise, effort, and time in reviewing our submission. Your invaluable feedback has greatly helped us improve our work. As the discussion has concluded, we summarize our claimed contributions and our responses to the reviewers’ concerns as follows.

---

**Contributions**:

* **Pioneering Perspective for Automatic Stage Light Control (ASLC)**: We propose a new way to view ASLC as a creative generation task, fundamentally different from previous rule-based methods that suffer from pattern monotony and low rationality. This perspective opens new avenues for future developments in ASLC. `1ux2` `PdJ6` `JUmG`

* **Novel and Robust Methodology**: To address the generation-based ASLC task, we introduce a novel framework, named Skip-BART, featuring a well-designed model structure and training process. Specifically, the innovative skip-connection module enhances the model's ability to capture the relationship between music and light at the frame level. `1ux2` `JUmG`

* **Promising Experimental Results**: Our experiments demonstrate the promising performance of Skip-BART, achieving levels comparable to human performance while significantly outperforming previous methods. Additionally, we have introduced the first ASLC dataset to support future research. `PdJ6` `JUmG`

---

**Concerns and Our Response**:

* **Lack of Pre-training Details**: We have included detailed model structure and training process information regarding pre-training in Appendix A, along with an ablation study to illustrate the efficacy of the random MASK and adversarial training strategy (Table 1). `1ux2` `JUmG`

* **Narrow Experimental Scope**: We acknowledge that the narrow style used in our experiments is a limitation. However, we have demonstrated the efficacy of our method through both in-domain and cross-domain experiments, utilizing the most common styles in livehouses. Further experimentation with other styles can be explored in future research, and we have addressed this in our Limitations and Future Work section. `1ux2`

* **Gap to Real Scenarios**: Consistent with most previous works, our current method focus only on single primary light control in an offline manner. As ASLC is still in its early stages, there are numerous challenges to address for online multiple light control, such as accurately predicting musical trends and adapting to heterogeneous stage lighting setups. We have noted this in our Limitations and Future Work section and believe that our innovative perspective on viewing ASLC as a creative generation task lays a solid foundation for this area. `1ux2` `JUmG`

* **Writing Improvement**: We have revised our Limitations and Future Work section, focusing on the gaps related to real-world multiple, online, and multi-style control, alongside potential technical directions. We also revised the introduction and human study setup sections for greater clarity. `PdJ6`

---

Finally, we express our gratitude once again for your contributions to this conference.

Sincerely,

ICLR 2026 Submission 4854 Authors

---

### Meta-Review · Area_Chair_himt · 2026-01-06

**Summary:**

The reviewer were positively impressed with the novelty of the problem and the technical solution. They noted that the paper opens a new research direction bridging music information retrieval and stage performance automation, with primary emphasis being conceptual innovation. Real-time light control remain a topic for future work, but since such application is not the topic of the current paper, this possible weakness should not affect adversely the recommendation to accept the paper in its current form.

**Reviewer Concerns:**

In the rebuttal stage the authors addressed the concerns about pre-training details, generalization abilities and the limited scope of the method focusing only on single primary light control in an offline manner.

**Reviewer Scores:**

Reviewer JUmG accepted the responses as being good quality. I believe the other reviewers would maintain their scores in view of the responses.

---

### Decision · Program_Chairs · 2026-01-26

Accept (Poster)